# PGC-1α mediates a metabolic host defense response in human airway epithelium during rhinovirus infections

Aubrey N. Michi [1,2], Bryan G. Yipp [2,3], Antoine Dufour [1,4], Fernando Lopes [5,6] & David Proud [1,2✉]

Human rhinoviruses (HRV) are common cold viruses associated with exacerbations of lower airways diseases. Although viral induced epithelial damage mediates inflammation, the molecular mechanisms responsible for airway epithelial damage and dysfunction remain undefined. Using experimental HRV infection studies in highly differentiated human bronchial epithelial cells grown at air-liquid interface (ALI), we examine the links between viral host defense, cellular metabolism, and epithelial barrier function. We observe that early HRV-C15 infection induces a transitory barrier-protective metabolic state characterized by glycolysis that ultimately becomes exhausted as the infection progresses and leads to cellular damage. Pharmacological promotion of glycolysis induces ROS-dependent upregulation of the mitochondrial metabolic regulator, peroxisome proliferator-activated receptor-γ coactivator 1α (PGC-1α), thereby restoring epithelial barrier function, improving viral defense, and attenuating disease pathology. Therefore, PGC-1α regulates a metabolic pathway essential to host defense that can be therapeutically targeted to rescue airway epithelial barrier dysfunction and potentially prevent severe respiratory complications or secondary bacterial infections.

[1] Department of Physiology and Pharmacology, Cumming School of Medicine, University of Calgary, Calgary, AB, Canada. [2] Snyder Institute for Chronic Diseases, University of Calgary, Calgary, AB, Canada. [3] Department of Critical Care Medicine, Cumming School of Medicine, University of Calgary, Calgary, AB, Canada. [4] McCaig Institute for Bone and Joint Health, University of Calgary, Calgary, AB, Canada. [5] Institute of Parasitology, McGill University, Ste-Anne-de-Bellevue, QC, Canada. [6] Department of Microbiology and Immunology, McGill University, Montreal, QC, Canada. ✉email: dproud@ucalgary.ca

Viruses hijack the transcriptional, translational, and packaging machinery of host cells to effectively replicate virions. As this machinery is diverted towards viral replication, it requires considerable energy to maintain cellular bioenergetic homeostasis and to initiate antiviral responses. As the majority of cellular energy production occurs by glycolysis, oxidative phosphorylation, and the citric acid (TCA) cycle, the stressed host cell reprioritizes ATP synthesis pathways to preserve important host defense functions (i.e., antiviral signaling, barrier function, cytokine release). Viruses have evolved diverse adaptations to redirect host cell metabolism to secure energy resources[1,2], yet our understanding of cellular metabolic responses that preserve essential functions during viral infection remains poorly understood.

Human rhinoviruses (HRVs) are the most common human respiratory viruses and are major triggers for acute exacerbations in individuals with chronic inflammatory lower airway diseases, such as asthma, chronic obstructive pulmonary disease, and cystic fibrosis[3–5]. There are over 150 strains classified into three species (HRV-A, -B, and -C) with diverse cellular receptor requirements (intercellular adhesion molecule-1 (ICAM-1), low-density lipoprotein receptor (LDLR), and cadherin-related family member 3 (CDHR3)[6]. The recently characterized HRV-C group of rhinoviruses are associated with more severe asthma exacerbations in children, particularly those expressing a CDHR3 allelic mutation[7]. Although it has been clearly established that HRVs induce airway epithelial cell production of antiviral host defense molecules and pro-inflammatory cytokine release[8–11], this investigation examines the mechanisms by which HRV-C infection alters airway epithelial cell host metabolism leading to epithelial barrier dysfunction.

HRV-C strains cannot be propagated in conventional cell lines or de-differentiated primary human airway epithelial cells as the receptor required for viral entry, CDHR3, is only functionally expressed on human airway ciliated cells[12]. Thus, we have performed the in vitro study entirely in highly differentiated cultures of human bronchial epithelial cells derived from human lung donors cultured at the air–liquid interface (ALI) (Supplementary Table 1) and corroborated our findings in nasal epithelial scrapings obtained during experimental in vivo HRV infections. This has necessitated the development of a number of novel methods to examine intracellular pathways in such differentiated cells.

In this work, we employ a quantitative proteomics and metabolomics approach to characterize virally altered metabolic pathways during HRV-C infection to explore the potential crosstalk between epithelial barrier preservation and changing metabolism. Our data indicate that HRV-C disrupts epithelial barrier function and shifts host cell metabolism into glycolysis marked by elevated lipid biosynthesis pathways, and efficient viral replication. Surprisingly, we find that by further shifting cell metabolism into glycolysis using the oxidative phosphorylation inhibitor, oligomycin A, epithelial barrier integrity is fully restored, and viral replication is largely diminished. We also discover that by fine-tuning the metabolic microenvironment by enhancing peroxisome proliferator-activated receptor-γ coactivator 1α (PGC-1α) activity, we can preserve barrier function, reduce viral replication, and rescue airway epithelial cells from severe HRV-C disease pathophysiology. To confirm the physiological relevance of this therapeutic target, we use data from two experimental human HRV infection studies in which we find that PGC-1α is downregulated in HRV-infected patients.

## Results

**HRV-C drives severe epithelial barrier loss in ALI cultures**. We used confocal microscopy to investigate the effects of equivalent infectious doses of HRV-16 (uses ICAM-1 receptor), HRV-1A (uses LDL receptor), and HRV-C15 (uses CDHR3 receptor) on tight junction organization (ZO-1 and occludin) at 24 h post infection in highly differentiated ALI cultures (Fig. 1a). HRV-C15 induced the greatest visual disruption of ZO-1 and occludin. These relative changes were corroborated by increased apical to basolateral permeability to the fluorescent tracer fluorescein isothiocyanate (FITC)-dextran (3–5 kDa) at 24 h post infection (Fig. 1b). Barrier function disruption was dependent on actively replicating the virus as it was not observed using replication-deficient ultraviolet light-treated HRV-C15 (Fig. 1c and Supplementary Fig. 1d). Strain-specific permeability effects were not due to viral replication differences, as all three strains yielded equivalent amounts of intracellular viral RNA (Supplementary Fig. 1a). Given that HRV-C15 induced the greatest amount of epithelial barrier loss and is associated with more profound exacerbations in asthmatic children, we further investigated the mechanisms underlying HRV-C15-induced barrier dysfunction. HRV-C15 replication kinetics were determined by infecting ALI cultures with $10^7$ and $10^9$ RNA copy number HRV-C15. Apical release was determined by serially washing the mucosal surface every 24 h and matched intracellular virus was quantified by collecting cell lysates for viral RNA extraction and quantitative reverse transcription-PCR (RT-qPCR) (Fig. 1d). For the HRV-C15 dose initially used for barrier function experiments ($10^9$ RNA copy number), both apical and intracellular virus levels peaked at 24 h post infection. Thus, we examined viral replication-associated events occurring at times prior to and including peak replication at 24 h using an infectious dose of $10^9$ RNA copy number.

The ALI epithelium is marked by extensive extrusion of infected ciliated cells at 24 h post infection, which stain positive for the viral RNA replication intermediate, double-stranded RNA (dsRNA), and show disorganization of tight junction proteins (i.e., ZO-1, occludin) (Fig. 1i). However, preceding this severe pathology, FITC-dextran leak was evident from 4 to 8 h post infection (Fig. 1f, top) in the absence of detectable cytotoxic events (Supplementary Fig. 1b). At these early timepoints, HRV-C15 titers were still relatively low (Fig. 1e), and transepithelial electrical resistance (TEER) remained unchanged (Fig. 1g). We speculated that HRV-C15 replication and downstream epithelial antiviral responses may be initiated in tandem with early barrier dysfunction. However, increases in messenger RNA (mRNA) levels for replication-dependent interferon-$\lambda_1$ (IFN-$\lambda_1$) (Fig. 1h, top) and IFN-β (Supplementary Fig. 1c) and the antiviral molecule viperin (Fig. 1h, bottom) were not observed until 16–20 h post infection. Since induction of these antiviral responses is thought to be dependent on dsRNA signaling, this implies that dsRNA-induced signaling occurs after the initiation of barrier loss, which is consistent with the lack of effect of the synthetic dsRNA, poly I:C, on barrier function (Fig. 1c). Although we cannot rule out that recognition of dsRNA occurs much earlier than interferon mRNA transcription, our data, taken together, suggest that this innate immune signaling pathway is not required for initiating early barrier loss.

**Proteomic analysis reveals changing metabolic and cytoskeletal pathways during HRV-C15 infection**. To investigate the potential metabolic mechanisms driving early barrier loss, we performed proteomic and metabolomic analyses of HRV-C15-infected ALI cultures derived from six individual healthy human lung donors. These analyses were performed in parallel on whole-cell lysates (proteomics) and basolateral media-secreted metabolites (metabolomics) at 4, 12, and 24 h post infection (Fig. 2a). We identified 2700 total proteins in the ALI cultures (Supplementary

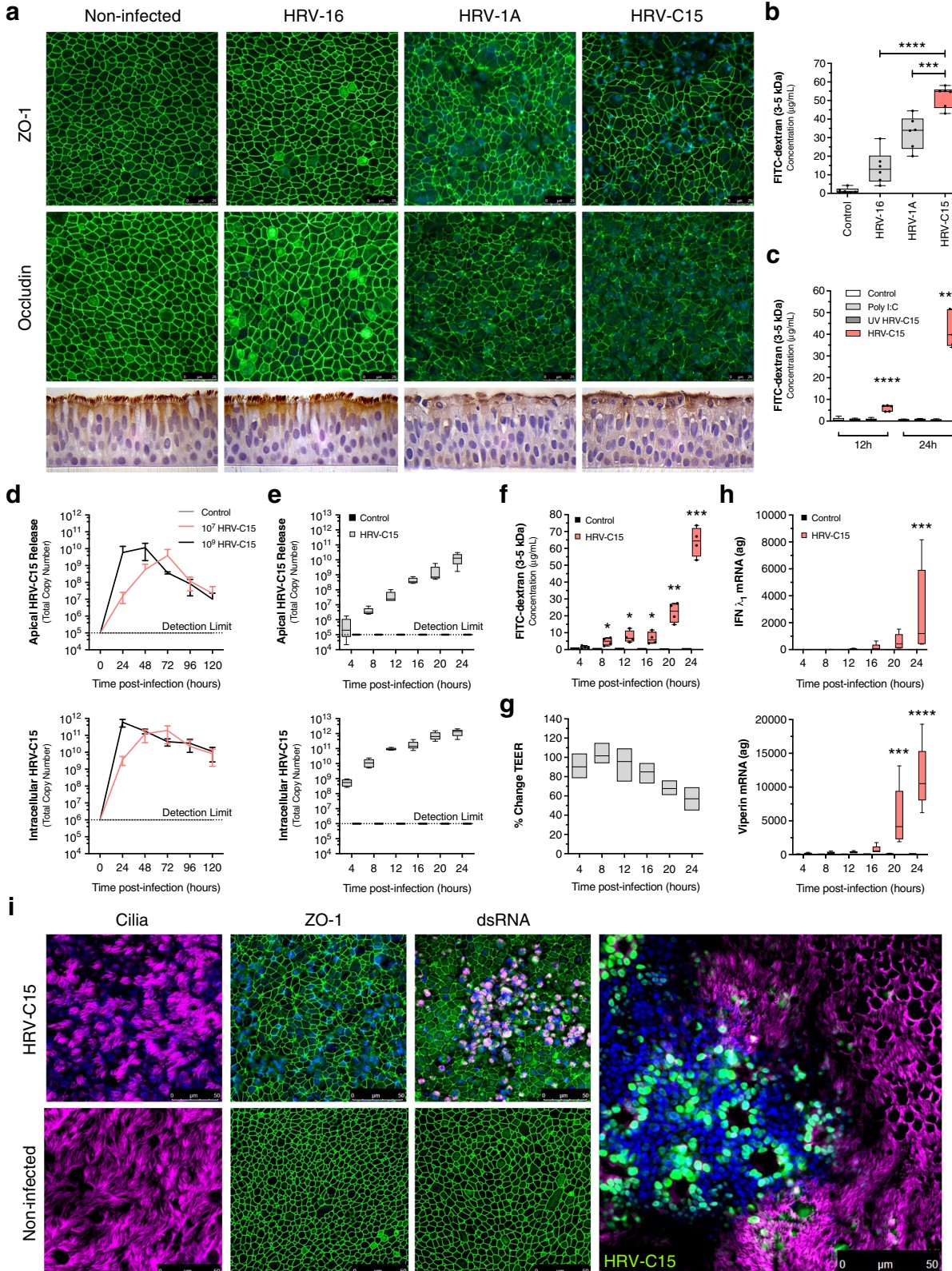

Data 1). Proteomics gene overlap analysis indicated that numerous genes at 4 h post-HRV infection also appear at 12 and 24 h post infection. However, genes appearing at 12 and 24 h become more unique to each time point (Fig. 2b). We speculate that the responses to HRV at 4 h may be reflective of generalized responses to pathogen infiltration and become more specialized as time progresses to virus-specific responses as the infection

progresses. Proteomic enrichment ontology cluster analysis (Fig. 2c) demonstrated that 4 h post-HRV infection is marked by enhanced epithelial signaling for proliferation (i.e., Notch pathways) and regulation of responses to the virus by a virus. We also observed signs of increased metabolic demands for energy, implicated by the enrichment of autophagy pathways (Fig. 2c). At 12 h post-HRV infection, the increased metabolic demands were

**Fig. 1 HRV-C drives epithelial barrier loss in air–liquid interface cultures. a**, Top: representative immunofluorescence images of HRV strain comparison of ZO-1 and occludin (green) and DAPI (blue) at 24 h post infection. Representative confocal images were obtained from four independent experiments performed using ALI cultures derived from four different lung donors and selected from ten fields per condition per donor. Scale bar, 25 μm. Bottom: representative histological sections of HRV strain comparison. Representative histological section images using a ×100 objective were obtained from four independent experiments performed using ALI cultures derived from four different lung donors and selected from ten fields per condition per donor, $n = 4$ donors. **b** Permeability assay measuring apical to basolateral FITC-dextran (3–5 kDa) flux at 24 h post infection with either HRV-16, HRV-1A, or HRV-C15 in ALI cultures. Data were analyzed by one-way ANOVA with Holm–Sidak multiple comparisons: ***$p = 0.0002$, ****$p < 0.0001$, $n = 6$ donors. **c** Permeability assay measuring FITC-dextran (3–5 kDa) flux at 12 and 24 h post exposure to live HRV-C15, poly I:C (30 μg/mL), or UV-treated HRV-C15. Data are represented as boxplots and analyzed by one-way ANOVA with Dunnett multiple comparisons to noninfected ALI cultures at each time point: ****$p < 0.0001$, $n = 5$ donors. **d** HRV-C15 replication kinetics in ALI cultures ($10^7$ and $10^9$ RNA copy number input dose) determined by serial apical DPBS washes (top) and intracellular lysates (bottom) collected at 24 h intervals for 5 days. There was no detectable HRV-C15 in noninfected ALI cultures. Dashed line represents RT-PCR detection limit. Data are represented as mean ± SD, $n = 3$ donors. **e** HRV-C15 replication kinetics in ALI cultures ($10^9$ RNA copy number input dose) determined by cumulative DPBS washes (top) and intracellular lysates (bottom) collected at 4 h intervals for 24 h. There was no detectable HRV-C15 in noninfected ALI cultures. Dashed line represents RT-PCR detection limit. Data are represented as boxplots, $n = 5$ donors. **f** FITC-dextran (3–5 kDa) flux in HRV-C15-infected ALI cultures during 24 h. Data are represented as boxplots in one figure for optimal visualization and analyzed by two-tailed paired *t* test to matched timepoint noninfected controls, from left to right: *$p = 0.034$, *$p = 0.027$, *$p = 0.040$, **$p = 0.005$, ***$p = 0.0007$, $n = 4$ donors. **g** Transepithelial electrical resistance (TEER) serially measured during HRV-C15 infection of ALI cultures over 24 h. Data are represented as minimum to maximum floating bars indicating mean (center line) of percent change of matched timepoint noninfected ALI cultures, $n = 3$ donors. **h** IFN-$\lambda_1$ (top) and viperin (bottom) mRNA expression during HRV-C15 infection in ALI cultures over 24 h, quantified by RT-PCR. There were no detectable levels of IFN-$\lambda_1$ or viperin in noninfected ALI cultures. Data are represented as boxplots and analyzed by one-way ANOVA with Dunnett multiple comparisons: IFN-$\lambda_1$ ***$p = 0.0003$, viperin ***$p = 0.0002$, ****$p < 0.0001$, $n = 5$ donors. **i** Representative immunofluorescence images of HRV-C15-infected ALI cultures at 24 h post infection indicating ZO-1 (green), cilia (pink), DAPI (blue), and dsRNA (pink), $n = 3$. Merged representative image (far right) represents HRV-C15 localization in the ALI epithelium: cilia (pink), DAPI (blue), and dsRNA (green). Representative confocal images were obtained from three independent experiments performed using ALI cultures derived from three different lung donors and selected from ten fields per condition per donor. Scale bar, 50 μm. Data represented as boxplots indicate the median (center line), upper and lower box bounds (IQR = first and third quartiles), and whiskers (min and max values), with individual donor data points superimposed onto the boxplot.

confirmed by pathway enrichment of glucose transporter type 4 (GLUT4) translocation to the plasma membrane and enhanced glycosphingolipid metabolism. Additional enriched pathways at 12 h post-HRV infection corroborate the loss of epithelial barrier function observed in Fig. 1f, such as regulation of the actin cytoskeleton by Rho-GTPases and intermediate filament cytoskeletal reorganization. At 24 h post-HRV infection, we found increased alcohol/lipid biosynthesis pathways and other metabolic processes, such as regulation of alcohol biosynthetic processes, glycosyl compound metabolic processes, primary alcohol metabolic processes, and initiation of IFN-α/β signaling (Fig. 2c). Enriched ontology from the full cluster (4, 12, and 24 h) was converted into a network layout to depict relevant pathways to HRV infections such as antiviral mechanism by IFN-stimulated genes, membrane trafficking, plasma membrane bounded cell projection assembly, regulation of plasma membrane bounded cell projection organization, positive regulation of cell motility, translocation of GLUT4 to the plasma membrane, response to xenobiotic stimulus, regulation of the actin cytoskeleton by Rho-GTPases, and autophagy (Fig. 2d). All protein–protein interactions (PPIs) among each time point gene list were extracted from the PPI data source and used to form PPI networks (Fig. 2e–h). GO enrichment analysis was applied to the networks to assign biological meanings and Molecular Complex Detection (MCODE) term descriptions are listed in Table 1. Interestingly, at 4 h post-HRV infection, we observed responses to xenobiotic stimulus, regulation of plasma membrane bounded cell projection organization, cytokine signaling in immune system, regulation of canonical Wnt signaling pathway, and canonical Wnt signaling pathway (MCODE1) (Fig. 2e). At 12 h post-HRV infection, we observed membrane trafficking, and vesicle-mediated transport (MCODE1) (Fig. 2f), while at 24 h post-HRV infection, we observed mitochondrial responses such as mitochondrial translation elongation, mitochondrial translation termination, and mitochondrial translation initiation (MCODE2) (Fig. 2g). The merged list of PPI highlighted regulation of insulin-like growth factor (IGF) transport and uptake by IGF-binding proteins

(IGFBPs) (MCODE4), translocation of GLUT4 to the plasma membrane (MCODE6), exocyst complex (MCODE6), and intraciliary transport involved in cilium assembly (MCODE9) (Fig. 2h).

## HRV-C15 shifts host cell metabolism towards fatty acid biosynthesis.

We further expanded on our proteomic analyses by performing metabolomics on basolaterally released metabolites from the matched samples at 4, 12, and 24 h post-HRV infection (Fig. 3a). We identified 31 known metabolites in the ALI cultures (Supplementary Data 2). Although secreted metabolite changes occurred as early as 4 h post infection (Supplementary Fig. 2a–c), these findings, along with the 12 h timepoint (Supplementary Fig. 2d–f), were not statistically significant. However, at 24 h post infection, HRV-C15 infection upregulated ten metabolites and downregulated 15 (Top 25) (Fig. 3b). Metabolomic analysis identified significant enrichment of multiple fatty acid biosynthesis pathways (Fig. 3c), which corroborated our proteomic observations, and reports that rhinoviruses modulate lipid metabolism as a means to remodel endoplasmic reticulum and Golgi membranes as viral replication sites[13,14]. An increased accumulation of 3-hydroxybutanoic acid and ethanolamine phosphate and decreased detection of L-serine and alpha-D-glucose were observed at 24 h after HRV infection (Fig. 3d). These metabolites are associated with phosphatidylcholine biosynthesis, sphingolipid metabolism, phosphatidylethanolamine biosynthesis, and fatty acid biosynthesis pathways, which are essential pathways for forming lipid bilayers. Integrated proteomic and metabolomic analysis revealed enriched pathways (based on *p* value and biological relevance) that were dominated by arginine biosynthesis, glutathione metabolism, and D-glutamine and D-glutamate metabolism at all timepoints (Fig. 3e). Interestingly, pantothenate and coenzyme A (CoA) biosynthesis also appeared to be enriched at 24 h, when secreted metabolites reached statistical significance. Pantothenate (i.e., vitamin B5) is the main precursor for the biosynthesis of CoA and carrier proteins that have a phospho-

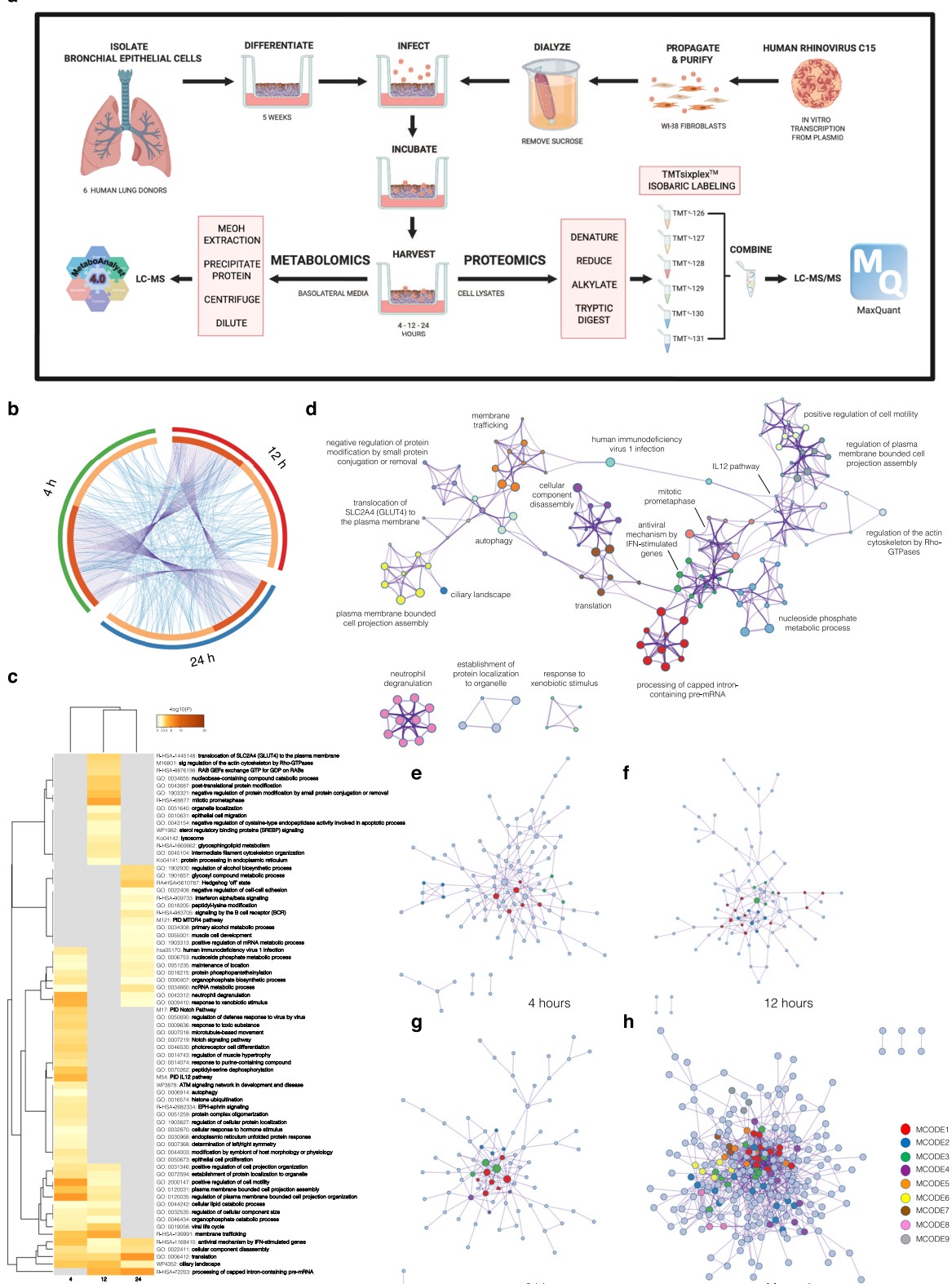

pantetheine prosthetic group. The phospho-pantetheine moiety is donated to these proteins by CoA and is used to shuttle intermediates between the active sites of enzymes involved in fatty acid, non-ribosomal peptide, and polyketide synthesis. CoA is an essential cofactor for cell growth and is involved in many metabolic reactions, including the synthesis of phospholipids, synthesis and degradation of fatty acids, and the TCA cycle.

The majority of fatty acid biosynthesis is derived from carbohydrates using glycolysis; therefore, we analyzed the transcriptional profiles of six metabolic-associated genes generally reflective of oxidative phosphorylation or glycolysis metabolic activity (Fig. 4a) during HRV-C15 infection. By 24 h post infection, hexokinase-2 (HK2), which is a rate-limiting enzyme that phosphorylates glucose to produce glucose-6-phosphate

**Fig. 2 Proteomic analysis reveals changing metabolic and cytoskeletal pathways during infection. a** Workflow schematic of proteomics and metabolomics experimental design. Graphic created using BioRender. **b** Circos plot indicating overlapping gene lists. Dark orange color represents genes that appear in multiple lists and light orange color represents genes that are unique to that gene list. Purple lines link the same gene that is shared by multiple gene lists. Blue lines link the different genes where they fall into the same ontology term (the term must be statistically significantly enriched and with size no >100). **c** Heatmap of statistically enriched terms in which accumulative hypergeometric p values and enrichment factors were calculated and used for filtering. The remaining significant terms were hierarchically clustered into a tree based on kappa-statistical similarities among their gene memberships. A 0.3 kappa score was applied as the threshold to cast the tree into term clusters. Dendrogram displays selected terms with the best p value within each cluster as its representative term. Heatmap cells are colored by their p values, and gray cells indicate the lack of enrichment for that term in the corresponding gene list. **d** Subset of representative terms from clusters in a network layout. Each term is represented by a circle node, in which the size is proportional to the number of input genes that fall into that term, and the color represents its cluster identity. Terms with a similarity score >0.3 are linked by a line (line thickness represents the similarity score). **e** Protein–protein interactions (PPIs) among each input gene list were extracted from PPI data source and formed a PPI network at each time point. Molecular Complex Detection (MCODE) term descriptions and associated p values are listed in Table 1. At 4 h post infection, MCODE1: R-HSA-1280215, GO:0060828, GO:0060070; MCODE2: R-HSA-6807878, R-HSA-199977, R-HSA-948021. **f** At 12 h post infection, MCODE1: R-HSA-68877, R-HSA-199977, R-HSA-68886; MCODE2: R-HSA-72163, R-HSA-72172, R-HSA-72203; MCODE3: R-HSA-8951664, GO:0043687. **g** At 24 h post infection, MCODE1: R-HSA-72163, R-HSA-72172, R-HSA-72203; MCODE2: R-HSA-5389840, R-HSA-5419276, R-HSA-5368286; MCODE3: R-HSA-8953854; MCODE4: R-HSA-72202, GO:0071427, GO:0006406. **h** Merged timepoint list MCODE1: R-HSA-72163, R-HSA-72172, R-HSA-72203; MCODE2: R-HSA-6807878, R-HSA-199977, R-HSA-948021; MCODE3: R-HSA-72766, GO:0006412; MCODE4: R-HSA-8957275, R-HSA-381426, GO:1901652; MCODE5: R-HSA-8951664, GO:0043687; MCODE6: R-HSA-1445148, CORUM:6167, CORUM:89; MCODE7: GO:0070268, R-HSA-6809371, R-HSA-6805567; MCODE9: GO:0035735, GO:0042073, R-HSA-5620924.

during glucose metabolism and is a key player in protective cell survival pathways and autophagy[15], was highly upregulated presumably to mitigate epithelial damage from HRV-C15 (Fig. 4a). Conversely, PGC-1α (gene name *PPARGC1A*), which is a key mitochondrial biogenesis transcriptional coactivator[16], is upregulated at 4 h post infection and acutely downregulated throughout infection (Fig. 4a). To corroborate the transcriptional induction of *PPARGC1A*, we confirmed by immunoblotting that PGC-1α protein was also elevated at 4 h post infection (Fig. 4b), implying that transcription and translation are tightly coupled for this gene, as may be expected given that PGC-1α is a dynamically controlled molecule. The analysis of post-translational modifications governing PGC-1α activity was not feasible due to the lack of reliable antibodies to acetylated and phosphorylated targets. We speculated that cellular conversion to glycolysis late in infection was supportive of HRV-C15 replication; therefore, we evaluated the role of glycolysis in regulating early barrier loss within the first 12 h of infection.

**Inhibiting ATP synthase improves barrier performance during HRV-C15 infection and lowers viral titers**. We observed that the same HRV strain-specific hierarchy in barrier loss (Fig. 1b) was reflected in similar *PPARGC1A* downregulation and *HK2* upregulation patterns (Fig. 4c). As HRV-C15 replication favors glycolysis associated with barrier loss, we postulated that shifting cell metabolism further into glycolysis would exacerbate HRV-C15-induced barrier loss. We pharmacologically shifted ALI culture metabolism towards glycolysis by inhibiting ATP synthase (mitochondrial complex V) using a widely accepted inhibitor of oxidative phosphorylation, oligomycin A[17,18]. Immediately following HRV-C15 infection, we administered oligomycin A and measured barrier function 12 h post infection by FITC-dextran movement. Contrary to what we expected, oligomycin A not only restored barrier function but also reduced HRV-C15 intracellular RNA copy number by 1.5–2 logs at 12 h post infection (Fig. 4d). Doses of oligomycin A alone in ALI cultures ranging from 3 to 30 μM had a marginal effect on epithelial barrier function, but when HRV-C15 was present, barrier restoration occurred at all doses; therefore, we chose the lowest effective dose (3 μM) for our experiments (Supplementary Fig. 3a). Interestingly, all doses of oligomycin A reduced viral replication with equivalent efficacy (Supplementary Fig. 3b), further uncoupling viral burden and barrier function. As inhibiting oxidative phosphorylation to favor glycolysis improved barrier function, we hypothesized that

inhibiting glycolysis with 2-deoxy-2-glucose (2-DG) would negatively affect barrier function. Unlike oligomycin A, 2-DG alone was sufficient to induce epithelial leak on par with HRV-C15 infection (Fig. 4e), indicating that ALI cultures require glycolysis to maintain tight epithelial barrier function. The barrier loss induced by 2-DG alone was so prominent that HRV-C15 infection presented no further worsening. If examining HRV titers alone, it would seem that inhibiting either oxidative phosphorylation or glycolysis would be antiviral, but the cost to the host cell of inhibiting glycolysis is a loss of the critical mucosal defense mechanism of barrier integrity, which could permit opportunistic secondary infections (Fig. 4f).

**Oligomycin A enhances oxygen consumption to fuel ROS production**. Mitochondrial metabolism and epithelial barrier crosstalk have not been studied in HRV infections of highly differentiated airway epithelial cell cultures. Although Unger et al. measured mitochondrial reactive oxygen species (ROS) in response to HRV infection[19], this work was performed using the 16HBE cell line, which, unlike primary epithelial cells, does not lead to a differentiated epithelium at ALI. Moreover, their examination of mitochondrial ROS was conducted at 16 h after infection, which is far later than when we see epithelial barrier alterations. Therefore, we measured ALI metabolic responses to oligomycin A using a Seahorse XFe24 Analyzer to measure real-time oxygen consumption rate (OCR) and extracellular acidification rate (ECAR). From our Seahorse metabolic studies using both non-airway epithelial cell lines (i.e., Caco-2) (Supplementary Fig. 4a) and primary bronchial epithelial cells (non-differentiated monolayers) (Supplementary Fig. 4b), inhibiting ATP synthase using oligomycin A decreases OCR within seconds as the cells are fixed in a short circuit of complex IV mitochondrial respiration. Surprisingly in ALI cultures, oligomycin A produced the opposite effect of enhancing OCR as measured by a Seahorse XFe24 Analyzer (Fig. 4g, top). This atypical response to inhibiting oxidative phosphorylation was validated by measuring ECAR (Fig. 4g, bottom), which indicated enhanced media acidification consistent with a glycolysis shift. We hypothesized that oxygen consumption, rather than fueling ATP production through oxidative phosphorylation, was redirected to generating ROS. Using CellROX, a fluorescent probe for confocal imaging live cell intracellular ROS, we found that oligomycin A induces ROS on a similar scale as 1 μM $H_2O_2$ (Fig. 4h). If oligomycin A directs the ALI cultures into ROS production, it is possible that ROS benefits

**Table 1 Protein–protein interaction MCODE descriptions.**

| Time | MCODE | Term | Description | Log 10(p) |
|---|---|---|---|---|
| 4 h | MCODE1 | R-HSA-1280215 | Cytokine signaling in immune system | −5.2 |
| | | GO: 0060828 | Regulation of canonical Wnt signaling pathway | −4.7 |
| | | GO: 0060070 | Canonical Wnt signaling pathway | −4.5 |
| | MCODE2 | R-HSA-6807878 | COPI-mediated anterograde transport | −9.8 |
| | | R-HSA-199977 | ER to Golgi anterograde transport | −9.1 |
| | | R-HSA-948021 | Transport to the Golgi and subsequent modification | −8.7 |
| 12 h | MCODE1 | R-HSA-68877 | Mitotic prometaphase | −7.6 |
| | | R-HSA-199977 | ER to Golgi anterograde transport | −6.2 |
| | | R-HSA-68886 | M phase | −6.1 |
| | MCODE2 | R-HSA-72163 | mRNA splicing—major pathway | −13.2 |
| | | R-HSA-72172 | mRNA splicing | −13.0 |
| | | R-HSA-72203 | Processing of capped intron-containing pre-mRNA | −12.4 |
| | MCODE3 | R-HSA-8951664 | Neddylation | −10.4 |
| | | GO: 0043687 | Post-translational protein modification | −4.7 |
| 24 h | MCODE1 | R-HSA-72163 | mRNA splicing—major pathway | −7.2 |
| | | R-HSA-72172 | mRNA splicing | −7.1 |
| | | R-HSA-72203 | Processing of capped intron-containing pre-mRNA | −6.7 |
| | MCODE2 | R-HSA-5389840 | Mitochondrial translation elongation | −10.1 |
| | | R-HSA-5419276 | Mitochondrial translation termination | −10.1 |
| | | R-HSA-5368286 | Mitochondrial translation initiation | −10.1 |
| | MCODE3 | R-HSA-8953854 | Metabolism of RNA | −4.9 |
| | MCODE4 | R-HSA-72202 | Transport of the mature transcript to cytoplasm | −7.6 |
| | | GO: 0071427 | mRNA-containing ribonucleoprotein complex export from nucleus | −7.2 |
| | | GO: 0006406 | mRNA export from nucleus | −7.2 |
| Merged | MCODE1 | R-HSA-72163 | mRNA splicing—major pathway | −28.6 |
| | | R-HSA-72172 | mRNA splicing | −28.4 |
| | | R-HSA-72203 | Processing of capped intron-containing pre-mRNA | −26.9 |
| | MCODE2 | R-HSA-6807878 | COPI-mediated anterograde transport | −11.5 |
| | | R-HSA-199977 | ER to Golgi anterograde transport | −10.4 |
| | | R-HSA-948021 | Transport to the Golgi and subsequent modification | −9.9 |
| | MCODE3 | R-HSA-72766 | Translation | −5.6 |
| | | CORUM:320 | 55S ribosome, mitochondrial | −5.6 |
| | | GO: 0006412 | Translation | −5.6 |
| | MCODE4 | R-HSA-8957275 | Post-translational protein phosphorylation | −7.8 |
| | | R-HSA-381426 | Regulation of insulin-like growth factor (IGF) transport and uptake by IGF-binding proteins (IGFBP) | −7.6 |
| | | GO: 1901652 | Response to peptide | −6.9 |
| | MCODE5 | R-HSA-8951664 | Neddylation | −9.6 |
| | | GO: 0043687 | Post-translational protein modification | −4.4 |
| | MCODE6 | R-HSA-1445148 | Translocation of SLC2A4 (GLUT4) to the plasma membrane | −13.0 |
| | | CORUM:6167 | Exocyst complex | −9.8 |
| | | CORUM:89 | Exocyst Sec6/8 complex | −9.8 |
| | MCODE7 | GO: 0070268 | Cornification | −7.2 |
| | | R-HSA-6809371 | Formation of the cornified envelope | −7.0 |
| | | R-HSA-6805567 | Keratinization | −6.4 |
| | MCODE9 | GO: 0035735 | Intraciliary transport involved in cilium assembly | −8.6 |
| | | GO: 0042073 | Intraciliary transport | −8.2 |
| | | R-HSA-5620924 | Intraflagellar transport | −8.2 |

Corresponding MCODE descriptions and terms from PPI networks at 4, 12, and 24 h post-HRV-C15 infection from Fig. 2e–h.

barrier function, which is contrary to other epithelial cell models, particularly gastrointestinal epithelial cells, where sustained ROS damages barrier function[20,21].

**HRV-C15 suppresses mitochondrial ROS during infection**. As early as 4 h post infection when HRV-C15 replication is in its early stages, we observed a surge in ROS production, which was visualized using CellROX (Fig. 5a). This initial ROS burst coincided with a spike in oxygen consumption (Fig. 5b) and may be a protective host response to infection. HRV-C15 quickly suppresses ROS by 6–8 h post infection, which temporally coincides with early epithelial barrier loss (Fig. 1f) and a time-dependent decrease in OCR (Fig. 5c). We postulated that HRV-C15 drives

epithelial barrier damage through a mechanism that inhibits barrier-protective ROS, and that oligomycin A treatment boosts ROS, tipping the balance towards host defense to restore barrier function. ROS can be of mitochondrial or cytoplasmic origin, both with differential cellular functions, so we investigated the intracellular source of oligomycin A-induced ROS. Using the mitochondrial ROS scavenger, MitoTEMPO, we found that scavenging mitochondrial ROS in the presence of ROS-booster oligomycin A, in a concentration-dependent manner, reverts epithelial barrier loss back to a leaky phenotype (Fig. 4d). Although the cells from individual ALI lung donors show variations in response magnitudes, all donors show enhanced permeability upon mitochondrial ROS scavenging. This ROS-dependent restoration of barrier function led us to investigate

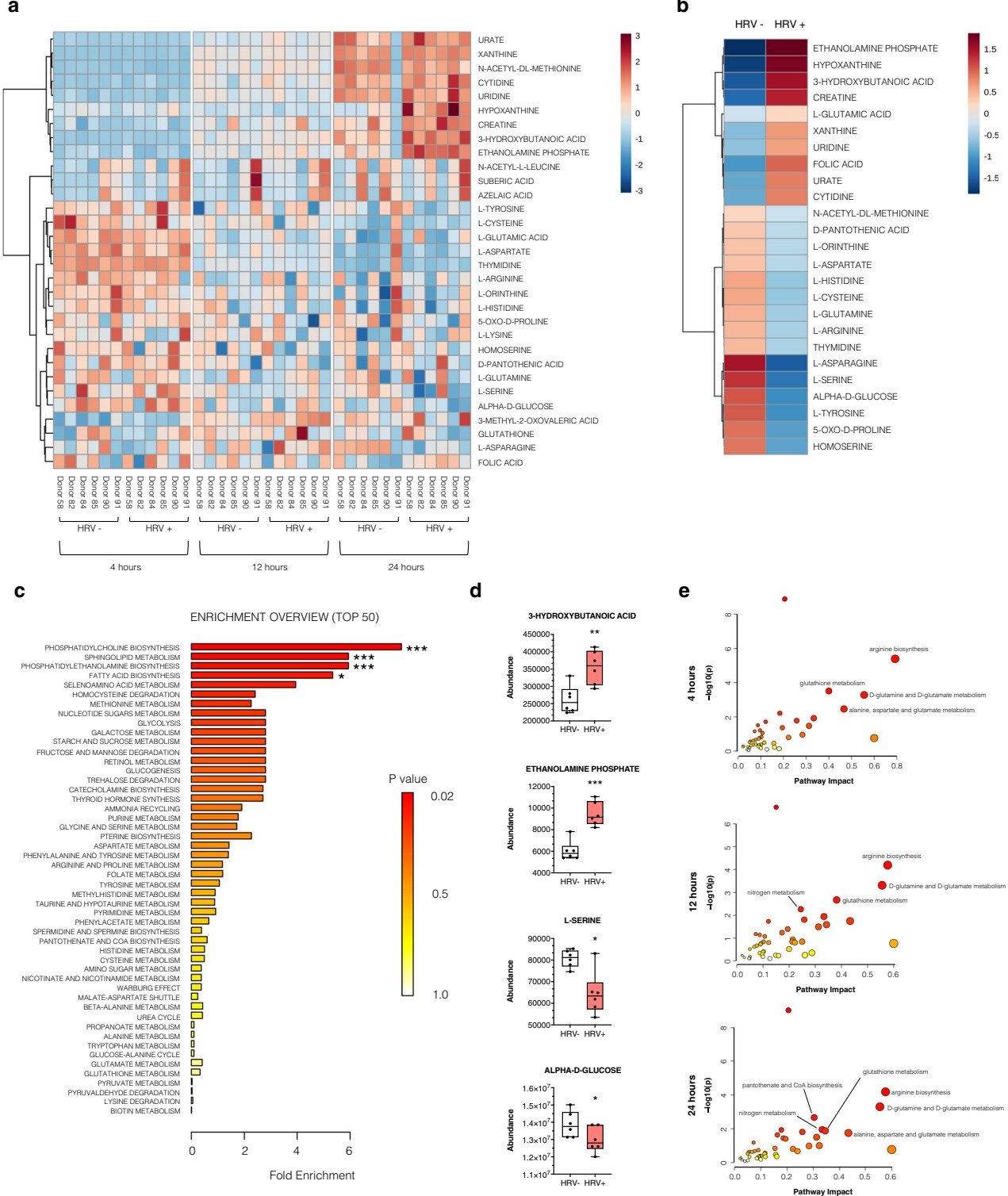

potential mechanisms by which ROS interacts with host cell metabolism to restore HRV-C15-induced barrier loss.

**Mitochondrial ROS drives PGC-1α activity to restore barrier function.** In our initial evaluation of the effect of HRV-C15 infection on six metabolic-associated genes, we found that levels of *HK2* (HK2) and *PPARGC1A* (PGC-1α) were most altered during HRV-C15 infection (Fig. 4a). While *HK2* was markedly upregulated in late infection where barrier dysfunction stems from epithelial damage and wounding, PGC-1α protein and

mRNA were induced during the 4-h post-infection window of elevated oxygen consumption, high ROS activity, and intact barrier function. Because HRV-C15 downregulates PGC-1α by 8–12 h post infection, we investigated the effect of oligomycin A on the expression of either *HK2* or *PPARGC1A* and observed that it markedly increases *PPARGC1A* expression (Fig. 5e). In light of this observation, we investigated the relationship between ROS and *PPARGC1A* expression and potential PGC-1α involvement in regulating barrier function. Scavenging mitochondrial ROS using MitoTEMPO lowers *PPARGC1A* mRNA expression (Fig. 5f)

**Fig. 3 HRV-C15 shifts host cell metabolism towards fatty acid biosynthesis. a** Hierarchical clustering heatmap depicting metabolite concentration (Top 25) of HRV-C15 (HRV+) and noninfected (HRV−) ALI cultures at 4, 12, and 24 h post infection normalized to noninfected control group at 4 h. Distance measure: Euclidean. Clustering algorithm: ward, $n = 6$ donors. **b** Heatmap depicting group averages of metabolite concentration (Top 25) of HRV-C15 (HRV+) and noninfected (HRV−) ALI cultures at 24 h post infection normalized to noninfected control group at 24 h. Distance measure: Euclidean. Clustering algorithm: ward, $n = 6$ donors. **c** Enrichment analysis of upregulated metabolites computed by Hits/Expected (hits = observed hits; expected = expected hits) of HRV-C15 (HRV+) and noninfected (HRV−) ALI cultures at 24 h post infection normalized to the HRV− group. Data were analyzed using MetaboAnalyst 4.0, Small Molecule Pathway Database (SMPDB) metabolite set library: phosphatidylcholine biosynthesis ***$p = 2.54e − 5$, sphingolipid metabolism ***$p = 3.02e − 4$, phosphatidylethanolamine biosynthesis ***$p = 3.02e − 4$, fatty acid biosynthesis *$p = 0.004$, $n = 6$ donors. **d** 3-Hydroxybutanoic acid, ethanolamine phosphate, L-serine, and alpha-D-glucose relative abundance during HRV-C15 infection in ALI cultures over 24 h determined by LC-MS. Data are represented as boxplots and analyzed by two-tailed paired $t$ test: 3-hydroxybutanoic acid **$p = 0.006$, ethanolamine phosphate ***$p = 0.0005$, L-serine *$p = 0.015$, alpha-D-glucose *$p = 0.023$, $n = 6$ donors. **e** Integrated metabolic pathway analysis on the data obtained from combined metabolomics and proteomics. Enrichment analysis by hypergeometric test and topology measured degree centrality. Combine queries integration method. The overviews show all matched pathways according to the $p$ values from the pathway enrichment analysis and pathway impact values from the pathway topology analysis. Data represented as boxplots indicate the median (center line), upper and lower box bounds (IQR = first and third quartiles), and whiskers (min and max values), with individual donor data points superimposed onto the boxplot.

without restoring HRV-C15 replication (Supplementary Fig. 4c). This corroborates evidence that ROS controls *PPARGC1A* expression and further dissociates the relationship between viral titer and barrier function. We next sought to pharmacologically alter this mitochondrial metabolic regulator to examine the relationship between PGC-1α activity and barrier function.

**PGC-1α restores barrier function and serves as a novel antiviral**. In the presence of a selective PGC-1α inhibitor, SR18292[22], full barrier restoration by oligomycin A was no longer possible at 12 h post infection (Fig. 6a). While *PPARGC1A* mRNA expression was attenuated by SR18292 (Fig. 6b), HRV-C15 titers remained unchanged at 12 h post infection (Fig. 6c). As oligomycin A triggers ROS production to enhance PGC-1α expression, we postulated that a PGC-1α activator alone may mimic the beneficial barrier-restorative effects of oligomycin A. Using the PGC-1α activator, ZLN005[23], we markedly improved HRV-C15-induced loss of barrier function in all lung donors (Fig. 6d) at 12 h post infection. While the use of a PGC-1α activator also increased transcription of *PPARGC1A* mRNA (Fig. 6e), it inhibited HRV-C15 replication (Fig. 6f), suggesting that manipulating PGC-1α activity may be a novel antiviral strategy with barrier-protective effects during rhinovirus infections.

**Stimulating early PGC-1α activity protects the host**. We hypothesized that metabolic intervention and modulating PGC-1α activity during early infection could rescue the severity of HRV-C15 driven epithelial barrier loss and associated tissue damage during advanced infection at 24 h. We intervened 2 h post infection with either oligomycin A or the PGC-1α activator, ZLN005 and we found that both ZLN005 and oligomycin A rescued HRV-C15-induced barrier loss at 24 h, albeit with differing efficacies (Fig. 7a). Oligomycin A stimulates greater PGC-1α activity, but ZLN005 also substantially reproduced the ability of oligomycin A to recover barrier and attenuate viral replication by 24 h post infection (Fig. 7c). While oligomycin A sustained *PPARGC1A* transcription at 24 h post infection, ZLN005 was unable to sustain high transcription levels, which may explain its less efficacious barrier recovery potential (Fig. 7b). In these matched experiments, we evaluated the visual distribution and organization of tight junction proteins ZO-1 (Fig. 7d) and occludin (Supplementary Fig. 5a) by confocal microscopy to show that tight junction disorganization and cell extrusion are largely prevented by both metabolic interventions.

**PGC-1α is a potential therapeutic target in clinical HRV infections**. In ALI cultures, multiple strains of HRV

downregulate PGC-1α during infection, with HRV-C15 displaying superior downregulation activity linked with more severe barrier loss (Figs. 1b and 4c). We corroborated this in vitro phenomenon using nasal epithelial cell scrapings obtained during two human experimental HRV infection studies. Because there is no HRV-C group preparation currently approved for human use, we used HRV-16 and HRV-39, which are currently approved for human use. We had previously published a randomized, parallel-group study with one group inoculated with HRV-16 ($n = 14$ patients) and the control group sham inoculated ($n = 16$ patients). Microarray analysis was performed on nasal scrapings taken 2 weeks prior to inoculation (baseline) and 48 h post-HRV or sham inoculation[8]. Analysis of our deposited dataset in Gene Expression Omnibus (GEO) (GSE11348) indicated downregulation of *PPARGC1A* at 48 h post-HRV-16 infection (Fig. 8a). To further confirm HRV downregulation of *PPARGC1A* by other clinically approved HRV strains, we used data from an ongoing experimental infection study infecting healthy volunteers with GMP grade HRV-39. In this study nasal epithelial scrapings were obtained before infection and at 48 and 72 h post infection, and RNA-sequencing (RNAseq) was performed ($n = 11$ patients). RNAseq analysis indicated that *PPARGC1A* is also downregulated in patients during HRV-39 infection (Fig. 8b).

## Discussion

Viral modulation of host cell metabolism is an emerging field[24,25] and the cellular metabolic responses to HRV infections warrant further investigations. To better understand the metabolic requirements of airway epithelial cells, highly differentiated ALI cultures are required to recapitulate the epithelial subtypes and architecture observed in vivo[26]. Consistent with previous studies, here we show that the bioenergetic profile and responses of ALI cultures to pharmacological interventions vary considerably from non-differentiated primary bronchial epithelial cultures, and even further from transformed or cancer cell lines (i.e., A549, BEAS-2B, etc.)[27,28]. Using pharmacological interventions, we demonstrated that cell metabolic intervention during HRV-C15 infection can preserve epithelial barrier function. Our studies produced three key findings: (1) metabolism regulates airway epithelial barrier function, (2) HRV-C15 impairs early barrier function prior to viral pattern recognition receptor driven responses through metabolic mechanisms, and (3) PGC-1α plays a key role in regulating barrier function and is a novel therapeutic target for recovering HRV-induced barrier loss while decreasing viral titers.

Epithelial barrier function is regulated by tight junction protein complexes that selectively control paracellular passage of macromolecules, ions, and water[29]. An intact mucosal epithelial

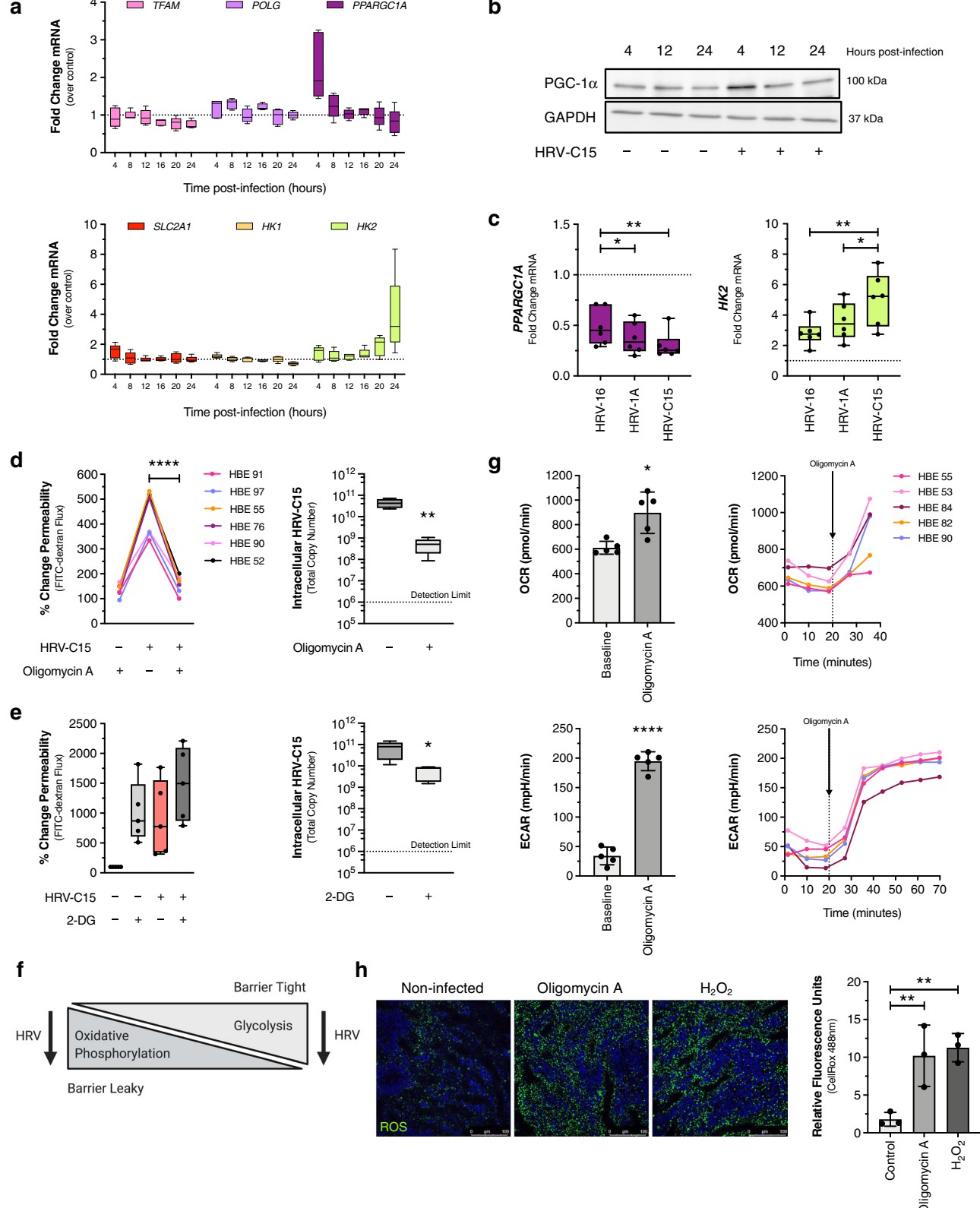

barrier prevents harmful intraluminal foreign antigens from entering the submucosa. There have been conflicting data on the effects of different HRV strains on epithelial barrier function. Although studies have shown reduced airway epithelial barrier function in response to HRV-1B[30,31] at 24 h after infection, no such changes were observed using HRV-16[32]. In the current study, we demonstrate that three different strains of HRV can

cause changes in epithelial barrier integrity. We confirm that HRV-16 caused the least loss of barrier function, perhaps due to the limited number of cells that can be infected by this strain[33], and that HRV-1A (closely related to HRV-1B) was more effective. However, we demonstrate that HRV-C15 was the most effective strain in inducing epithelial barrier changes, with major disruption in junctional architecture evident 24 h after infection. Thus

**Fig. 4 Metabolism drives barrier function. a** Metabolism-associated gene profiling of HRV-C15 infected ALI cultures over 24 h, assessed by RT-PCR. Data are represented as boxplots, $n = 5$ donors. **b** Representative immunoblot of ALI culture whole-cell lysates evaluating PGC-1α protein during HRV-C15 infection >24 h. GAPDH was used as a loading control. Representative immunoblot was selected from three independent experiments performed in ALI cultures derived from three different lung donors, $n = 3$ donors. **c** *PPARGC1A* and *HK2* mRNA expression by three strains of HRV at 24 h post infection. Data are represented as boxplots and analyzed by one-way ANOVA with Holm–Sidak multiple comparisons: *PPARGC1A* \*$p = 0.044$, \*\*$p = 0.004$, *HK2* \*$p = 0.049$, \*\*$p = 0.008$, $n = 6$ donors. **d**, Left: barrier function permeability assay using FITC-dextran (3–5 kDa) flux 12 h post infection in ALI cultures treated with or without 3 μM oligomycin A. Data are represented by individual human bronchial epithelial (HBE) donor lines and analyzed by one-way ANOVA with Holm–Sidak multiple comparisons: \*\*\*\*$p < 0.0001$, $n = 6$ donors. Right: matched intracellular HRV-C15 RNA copy number at 12 h post infection, quantified by RT-PCR. Dashed line represents RT-PCR detection limit. There was no detectable HRV-C15 in noninfected ALI cultures. Data are represented as boxplots and analyzed by two-tailed paired *t* test: \*\*$p = 0.003$, $n = 6$ donors. **e**, Left: barrier function permeability assay of FITC-dextran (3–5 kDa) flux 12 h post infection in ALI cultures treated with or without 100 μM 2-DG. Data are represented as boxplots and analyzed by one-way ANOVA. 2-DG treatment was not significantly different from HRV-C15 alone, $n = 5$ donors. Right: matched intracellular HRV-C15 RNA copy number at 12 h post infection, quantified by RT-PCR. Dashed line represents RT-PCR detection limit. There was no detectable HRV-C15 in noninfected ALI cultures. Data are represented as boxplots and analyzed by two-tailed paired *t* test: \*$p = 0.04$, $n = 6$ donors. **f** Graphical representation of metabolic pathways and HRV-C15 replication inhibition to summarize findings from (**d**, **e**). **g** Noninfected ALI cultures were biopsy punched (4 mm) and loaded into Seahorse XFe24 plates and OCR (top) and ECAR (bottom) measurements were performed in technical triplicates per donor prior to and following injection of 3 μM oligomycin A. Baseline OCR and ECAR were determined by the final reading prior to oligomycin injection compared to the maximal reading post-oligomycin injection. Real-time data are graphically represented by individual HBE donor lines, with a dashed line indicating oligomycin A injection event. Baseline OCR and ECAR bar graphs indicate mean ± SD and analyzed by paired *t* test: OCR \*$p = 0.012$, ECAR \*\*\*\*$p < 0.0001$, $n = 5$ donors. **h**, Left: representative immunofluorescence image of oxidative stress (ROS) in live ALI cultures detected using CellROX (green) and Hoechst 33342 (blue) following a 15-min incubation with either 3 μM oligomycin A or 1 μM $H_2O_2$. Scale bar, 100 μm. Representative confocal images and quantification were obtained from three independent experiments performed in ALI cultures derived from three different lung donors and selected from five fields per condition per donor. Data are represented as mean of five fields of view per donor ± SD and analyzed by one-way ANOVA with Holm–Sidak multiple comparisons: left to right \*\*$p = 0.008$, \*\*$p = 0.007$, $n = 3$ donors. Data represented as boxplots indicate the median (center line), upper and lower box bounds (IQR = first and third quartiles), and whiskers (min and max values), with individual donor data points superimposed onto the boxplot.

far, studies that have looked at potential mechanisms involved in HRV-induced barrier changes have focused on pattern recognition receptor-driven events at, or close to, 24 h after infection, often in cell lines. By this time point, it is clear that the epithelium is significantly compromised and that cells are being extruded from the culture. We performed time-course studies that clearly document that permeability changes are induced as early as 6–8 h post infection, well before pattern recognition receptor-mediated induction of antiviral genes is observed.

Much of our understanding of epithelial barrier function is derived from intestinal epithelial models, in which cytoskeletal signaling involving Rho-GTPases, calcium flux, and myosin light-chain phosphorylation mobilizes scaffolding proteins to regulate claudin and occludin organization[34–36]. Although these contractility processes also regulate the airway epithelial barrier, recent intestinal epithelial barrier function studies have shown impaired epithelial barrier function in metabolic disorders such as obesity and type 2 diabetes. Gastrointestinal barrier dysfunction in obesity is associated with enhanced microbial translocation into the lamina propria, primarily orchestrated by hyperglycemia-induced changes in the tight junction transcriptome[37]. Aside from metabolic disorders, HIV patients prescribed antiretroviral therapy (ART) have well-characterized non-AIDS-associated co-morbidities, one of which includes leaky gut. One study using simian immunodeficiency virus-infected primates and human immunodeficiency virus (HIV) patients found that HIV alters intestinal epithelial fatty acid metabolism and impairs mito-chondrial fatty acid β-oxidation, resulting in barrier dysfunction, and is further exacerbated by ART[38]. However, intestinal epithelial cell metabolism is continually reprogrammed by an abundance of microbial-derived butyrate, shifting the cells into hypoxia-inducible factor-regulated short-chain fatty acid meta-bolism, which promotes epithelial barrier homeostasis[39]. As the human airways are not hypoxic and have magnitudes less microbial burden, it is unlikely that airway epithelial metabolism is regulated in this manner.

Only one study to date has assessed highly differentiated nasal epithelial cell bioenergetic profiles. Although this study provided

valuable insight into airway epithelial glucose metabolism, it focused solely on glucose availability[28]. The current study investigates the relationship between cellular metabolism and barrier function in airway epithelial cells. Although a detailed longitudinal and integrated omics analysis was not performed for this cell biology-based study, both proteomic and metabolomic analyses clearly demonstrated that HRV infection created an increased metabolic demand for energy in epithelial cells and demonstrated activation of a number of biosynthetic pathways including fatty acid biosynthesis. We discovered that shifting ALI cultures preferentially towards either glycolysis or oxidative phosphorylation has marked effects on barrier function, espe-cially in HRV-C15-infected cells. However, we still do not understand how airway epithelial cells allocate ATP to preserve cellular functions when they are stressed or infected. There may be a dynamic balance between ATP production to maintain epithelial barrier function, to preserve innate immunological functions or even to support viral replication. When we inhibited oxidative phosphorylation, we may have biased the cells to a pathway of more rapid ATP production (glycolysis), but, as the barrier was restored, HRV replication was reduced. In cells where oxidative phosphorylation dominated, a slower mode of ATP production, barrier function was destroyed. Although the cells naturally revert to glycolysis during HRV-C15 infection, pre-sumably the host-protective shift is neither far enough nor fast enough as the cells show signs of barrier dysfunction in as little as 6–8 h post infection. When ALI cultures are exposed to inter-ventions that shift the cells into glycolysis earlier than they naturally drift while infected, we allow host metabolism to out-pace HRV-C15 replication to rescue barrier loss. We speculate that as cytoskeletal-associated tight junctions are altered early in infection, there may be mechanical cues by changes in the acto-myosin cytoskeleton that regulate glycolysis pathways[40]. Further investigations of specific cytoskeletal signaling pathways and metabolic pathways in the context of rhinovirus infections are needed.

PGC-1α is the most well-described member of the PGC-1 family. It is known to be a positive regulator of mitochondrial

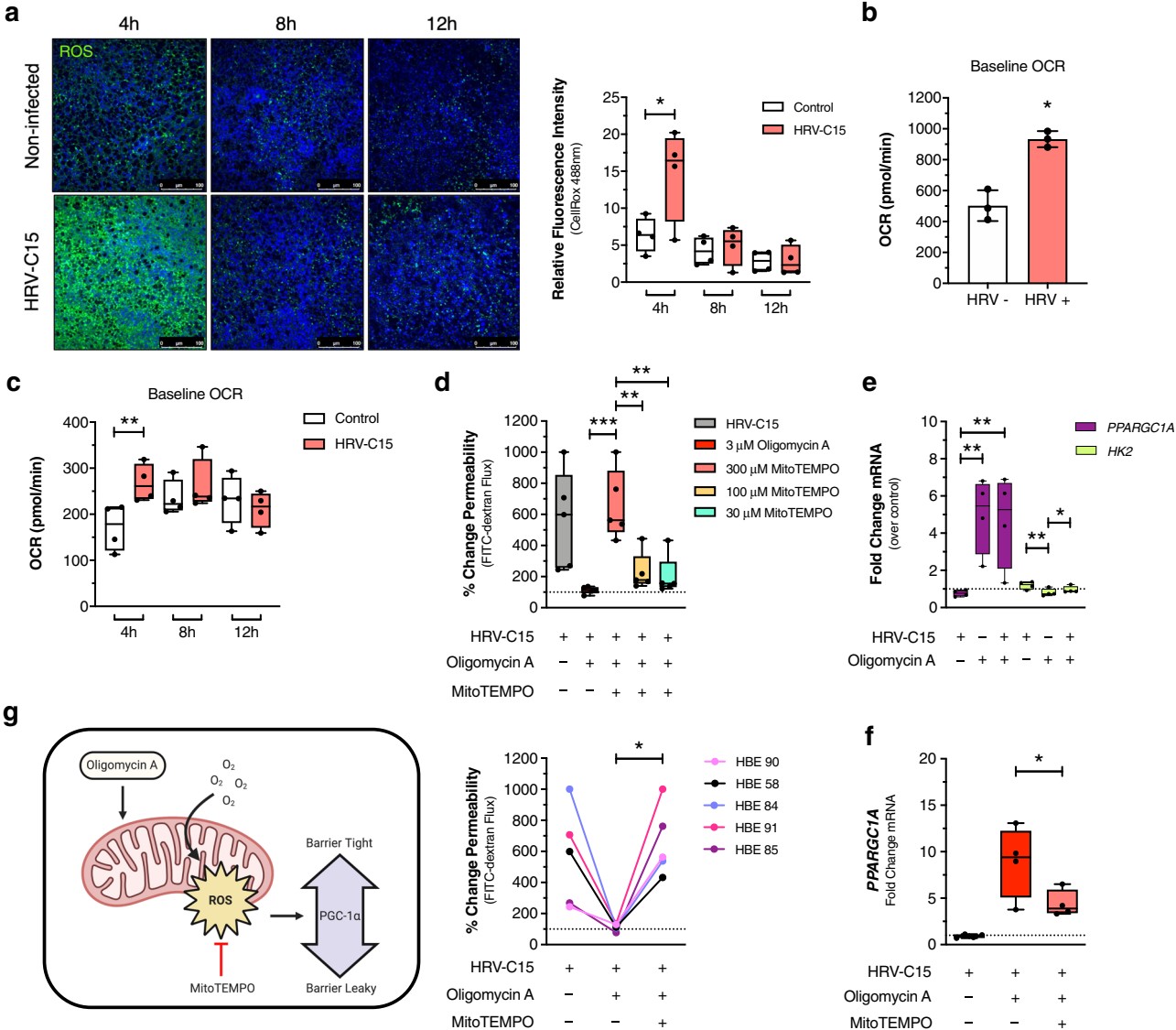

**Fig. 5 Oxygen consumption fuels mitochondrial ROS to restore barrier function. a**, Left: representative images of ROS in live HRV-C15-infected ALI cultures over 12 h, detected by CellROX (green) and nuclei by Hoechst 33342 (blue). Scale bar, 100 μm. Right: Quantification of CellROX fluorescence. Representative confocal images and quantification were obtained from four independent experiments performed in ALI cultures derived from four different lung donors and selected from three fields per condition per donor. Data are represented as boxplots of means of three fields of view per donor per condition and analyzed by two-way ANOVA with Sidak's multiple comparisons: **$p = 0.005$, $n = 4$ donors. **b** ALI cultures were infected with HRV-C15 or mock for 4 h, biopsy punched (4 mm), loaded into Seahorse XFe24 plates and baseline OCR was measured for three cycles. Individual donor points represent a mean of three technical replicates (biopsy punches) per donor per condition ± SD and analyzed by two-tailed paired $t$ test: *$p = 0.039$, $n = 3$ donors. **c** ALI cultures were infected with HRV-C15 or mock to perform Seahorse measurements on all matched timepoints simultaneously using staggered infections. ALI inserts were biopsy punched (2 mm), loaded into Seahorse XFe24 plates, and baseline OCR was recorded for four cycles. Data are represented as boxplots in one figure for optimal visualization comparison. Individual donor points represent the mean of four technical replicate punches per donor (four biopsy punches per condition per time point per donor) and analyzed by two-tailed paired $t$ test: **$p = 0.04$, $n = 4$ donors. **d** Top: MitoTEMPO (mitochondrial ROS scavenger) dose response in HRV-C15-infected ALI cultures treated with 3 μM oligomycin A, and permeability assessed by FITC-dextran (3–5 kDa) flux at 12 h post infection. Data are represented as boxplots and analyzed by one-way ANOVA with Holm–Sidak multiple comparisons: left to right ***$p = 0.0003$, **$p = 0.002$, **$p = 0.001$, $n = 5$ donors. Bottom: FITC-dextran (3–5 kDa) permeability of 300 μM MitoTEMPO barrier loss from oligomycin A restoration at 12 h post infection. Data are represented as individual HBE donor lines and analyzed by one-way ANOVA with Holm–Sidak multiple comparisons: *$p = 0.015$, $n = 5$ donors. **e** *PPARGC1A* and *HK2* mRNA expression affected by 3 μM oligomycin A treatment at 12 h post infection. Matched samples represented as boxplots in the same figure for optimal visualization comparison with each gene analyzed independently by one-way ANOVA with Holm–Sidak multiple comparisons: *PPARGC1A* **$p = 0.008$, *$p = 0.008$, *HK2* **$p < 0.006$, *$p = 0.039$, $n = 4$ donors. **f** *PPARGC1A* mRNA expression affected by 3 μM oligomycin A with or without 300 μM MitoTEMPO. Data are represented as boxplots and analyzed by one-way ANOVA with Dunnett multiple comparisons: *$p = 0.04$, $n = 4$ donors. **g** Graphical representation of mitochondrial ROS involvement in PGC-1α expression and barrier function to summarize findings from (**a**–**f**). Graphic created using BioRender. Data represented as boxplots indicate the median (center line), upper and lower box bounds (IQR = first and third quartiles), and whiskers (min and max values), with individual donor data points superimposed onto the boxplot.

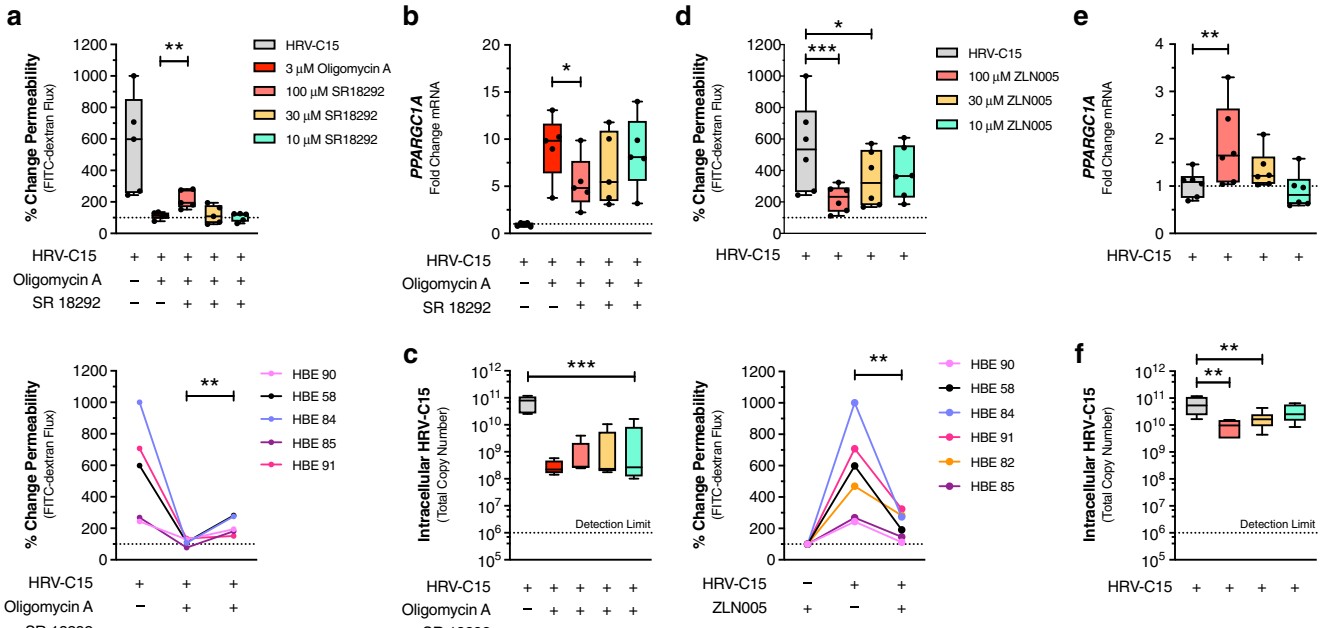

**Fig. 6 PGC-1α restores barrier function and serves as a novel antiviral. a** SR18292 (PGC-1α inhibitor) dose response on oligomycin A-treated ALI cultures measuring FITC-dextran (3–5 kDa) flux 12 h post infection (top). Loss of oligomycin A barrier restoration by 100 µM SR18292 (bottom). Data are represented as boxplots and individual HBE donor lines and analyzed by one-way ANOVA with Dunnett multiple comparisons: **$p = 0.005$, $n = 5$ donors. **b** *PPARGC1A* mRNA expression by treatment with 3 µM oligomycin A in the presence of SR18292 at 12 h post infection, measured by RT-PCR. Data are represented as boxplots and analyzed by one-way ANOVA with Dunnett multiple comparisons: *$p = 0.02$, $n = 5$ donors. **c** Matched HRV-C15 intracellular RNA copy number at 12 h post infection, quantified by RT-PCR. Dashed line represents RT-PCR detection limit. There was no detectable HRV-C15 in noninfected ALI cultures. Data are represented as boxplots and analyzed by one-way ANOVA with Dunnett multiple comparisons: ***$p = 0.0002$, $n = 5$ donors. **d** ZLN005 (PGC-1α activator) dose response in HRV-C15-infected ALI cultures measuring FITC-dextran (3–5 kDa) flux at 12 h post infection (top). HRV-C15 barrier recovery by 100 µM ZLN005 at 12 h post infection (bottom). Data represented as boxplots and individual donor lines and analyzed by one-way ANOVA with Holm–Sidak multiple comparisons: ***$p = 0.0008$, **$p = 0.03$, $n = 6$ donors. **e** *PPARGC1A* mRNA expression in response to ZLN005 treatment, measured at 12 h post infection. Data are represented as boxplots and analyzed by one-way ANOVA with Dunnett multiple comparisons: **$p = 0.002$, $n = 6$ donors. **f** Matched HRV-C15 intracellular RNA copy number, quantified by RT-PCR. Dashed line represents RT-PCR detection limit. There was no detectable HRV-C15 in noninfected cultures. Data are represented as boxplots and analyzed by one-way ANOVA with Holm–Sidak multiple comparisons: **$p = 0.001$, **$p = 0.005$, $n = 6$ donors. Data represented as boxplots indicate the median (center line), upper and lower box bounds (IQR = first and third quartiles), and whiskers (min and max values), with individual donor data points superimposed onto the boxplot.

biogenesis and respiration[41], antioxidant activity, peroxisomal remodeling[42], gluconeogenesis[43], and thermogenesis[44], and also plays a critical role in activating antioxidant enzymes to suppress ROS[45], by enhancing mitochondrial oxidative metabolism homeostasis. We demonstrated that three different strains of HRV (HRV-16, HRV-1A, and HRV-C15) all decreased epithelial expression of PGC-1α, showing the same rank order effect as was observed on disruption of barrier function. A key finding from these studies was that PGC-1α plays a critical role in the regulation of barrier function. Not only was epithelial barrier restoration by oligomycin A associated with increased expression of PGC-1α but the partial reversal of this effect by MitoTEMPO was associated with a partial reduction in PGC-1α. Moreover, alterations of the activity of PGC-1α using selective activator and inhibitor compounds also modulated barrier function as expected. Finally, the observation that sustaining increased expression of *PPARGC1A* beginning at 2 h post infection using ZLN005 or oligomycin A abrogated the marked disruption of epithelial architecture normally induced by HRV-C15 at 24 h post infection. Although there have been a few studies in intestinal epithelial cells investigating the relationship between PGC-1α and epithelial barrier function[38,46], to our knowledge, the role of PGC-1α in regulating airway epithelial cell homeostasis is unknown, particularly in the context of respiratory viral infections and metabolic stress. We used data from two experimental

HRV infection studies in healthy humans, one study using HRV-16 and microarray analysis of nasal scrapings 48 h after infection, and another study using HRV-39 and RNAseq of nasal scrapings 48 and 72 h after infection. Although the mechanistic insights into PGC-1α regulation of barrier function were performed in ALI cultures derived from bronchial epithelial cells, there is strong evidence to support that nasal epithelial cells largely recapitulate gene expression changes observed in bronchial epithelial cells as they share a 90% transcriptional overlap[47]. We also have previously found excellent agreement in HRV-induced gene expression profiles between nasal samples and in vitro cultured bronchial epithelial cells[8,48]. In both clinical studies, HRV-16 and HRV-39 considerably downregulated *PPARGC1A* during infection. This *PPARGC1A* downregulation was strongly recapitulated in ALI cultures, with HRV-C15 inducing the most robust downregulation activity of the strains tested. Given that promoting PGC-1α expression restores epithelial barrier function during HRV-C15 infection and suppresses viral replication in ALI cultures, PGC-1α may be a suitable target for therapeutic interventions to regulate responses to HRV-C15 infection in patients with lower airway diseases. Thus far, we have only performed studies demonstrating that modulation of PGC-1α activity regulated barrier function using infection with HRV-C15. Further studies are needed to extend this to additional strains, but the observation that multiple HRV strains downregulate PGC-1α

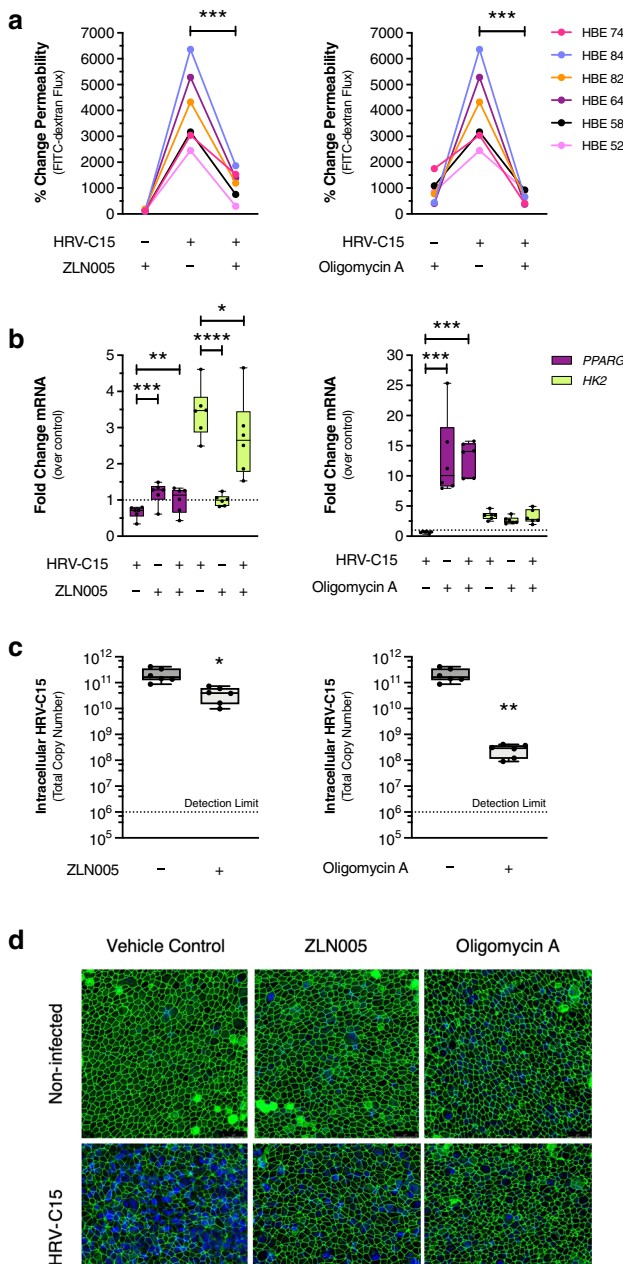

**Fig. 7 Stimulating early PGC-1α activity protects the host. a** FITC-dextran (3–5 kDa) permeability assay indicating barrier function recovery at 24 h post infection from early administration of 100 µM ZLN005 (left) or 3 µM oligomycin A (right). Data are represented as individual HBE donor lines and analyzed by one-way ANOVA with Holm–Sidak multiple comparisons: ZLN005 ***$p = 0.0002$, oligomycin A ***$p = 0.0006$, $n = 6$ donors. **b** Effects of 100 µM ZLN005 (left) or 3 µM oligomycin A (right) on *PPARGC1A* and *HK2* mRNA expression from matched samples derived from (**a**) at 24 h post infection. Data are represented as boxplots and *PPARGC1A* and *HK2* analyzed separately by one-way ANOVA with Holm–Sidak multiple comparisons. ZLN005: *PPARGC1A* ***$p = 0.0006$, **$p = 0.009$, *HK2* ****$p < 0.0001$, *$p = 0.036$. Oligomycin A: *PPARGC1A* ***$p = 0.0005$, ***$p = 0.005$ *HK2* not significant, $n = 6$ donors. **c** Matched intracellular HRV-C15 RNA copy number from ZLN005 or oligomycin A-treated ALI cultures from (**a**) at 24 h post infection, quantified by RT-PCR. Dashed line represents RT-PCR detection limit. There was no detectable HRV-C15 in noninfected ALI cultures. Data are represented as boxplots and analyzed by two-tailed paired *t* test: ZLN005 *$p = 0.011$, oligomycin A **$p = 0.009$, $n = 6$ donors. **d** Representative immunofluorescence images of ZO-1 (green) and nuclei (blue) from a matched experiment in (**a**) at 24 h post infection. Representative confocal images were obtained from three independent experiments performed using ALI cultures derived from three different lung donors and selected from ten fields per condition per donor, $n = 3$ donors. Scale bar, 25 µm. Data represented as boxplots indicate the median (center line), upper and lower box bounds (IQR = first and third quartiles), and whiskers (min and max values), with individual donor data points superimposed onto the boxplot.

and PGC-1α expression is a more rational target for therapeutic intervention that could be particularly beneficial in children with asthma or cystic fibrosis during HRV-C infections.

## Methods

**Materials**. A comprehensive list of resources, reagents, key equipment, and primer/probe sequences obtained from commercial vendors are provided in Supplementary Tables 2 and 3.

**Isolation of human bronchial epithelial cells**. Non-transplanted healthy human lungs were obtained using a tissue retrieval service (International Institute for the Advancement of Medicine, Edison, NJ). Organs were obtained from multiple hospitals across the United States. Informed consent for organs to be harvested for transplant or for research was obtained from family members at each hospital. Ethics approval to receive and use human lung tissues was obtained from the Conjoint Health Research Ethics Board of the University of Calgary (Ethics ID: REN15-0336) and the Internal Ethics Board of Institute for the Advancement of Medicine. No personal identifying information was provided for any of the donors. Because we receive tissue from anonymized donors after death and have no involvement with patients or families, no additional informed consent is required at our institution.

Human lungs were received 24–48 h post-surgical removal and dissected to isolate down to the third- and fourth-generation bronchial airways. Bronchial segments were incubated in 1 mg/mL Pronase in F12 media supplemented with gentamicin (50 µg/mL) for 36–40 h at 4 °C. Subsequently, bronchial segments were removed from the Pronase solution, placed in sterile Petri dishes, and further dissected to expose the luminal surface. The luminal surface was jetted using 10% fetal bovine serum (FBS) in F12 media using a 5 mL syringe upwards of 50 times, until luminal epithelial sloughing ceased. The epithelial cell containing FBS/F12 media were centrifuged at $311 \times g$ for 8 min at 4 °C[50]. Pelleted HBE cells were resuspended in Leibovitz's L15 media containing 20% FBS, 2% HEPES, 2% penicillin/streptomycin (2× P/S), and 15% dimethyl sulfoxide (DMSO). For cryopreservation, HBE cells were diluted to a final concentration of $0.5 \times 10^6$ cells/mL in a 1:1 mixture of 2× P/S and 2× DMSO (Leibovitz's L15 media containing 20% FBS, 2% HEPES, 2× P/S, and 15% DMSO) in 1 mL cryovial aliquots. HBE cells were stored at −80 °C for 24 h and transferred to liquid nitrogen for long-term storage.

**ALI culture of HBE cells**. One vial of HBE cells from one lung donor was considered one biological replicate ($n = 1$) for all experiments. HBE cells used to generate ALI cultures were expanded only once and not repeatedly passaged prior to seeding for ALI differentiation. HBE cells were thawed into T75 cm² flasks in

expression in the same rank order of effect as they modulate barrier function supports the concept that this pathway may be of general importance for multiple HRV strains.

Antiviral therapies typically aim to inhibit viral binding or suppress viral replication, ultimately reducing viral titers. However, levels of viral shedding alone may not adequately reflect effects on airway cell function. For example, Gualdoni et al. inhibited glycolysis using 2-DG during HRV replication in fibroblasts and HeLa cells and reported that it reduced viral titers, leading the authors to speculate that inhibiting glycolysis held the therapeutic potential for HRV infections[49]. While our studies here confirm that inhibiting glycolysis lowers HRV titers in highly differentiated airway epithelial cells, this leads to drastic impairment of epithelial barrier function, increasing the potential for opportunistic lung infections. Thus, while both oligomycin A and 2-DG can suppress viral titers, differential effects on epithelial function suggest that the regulation of oxidative phosphorylation

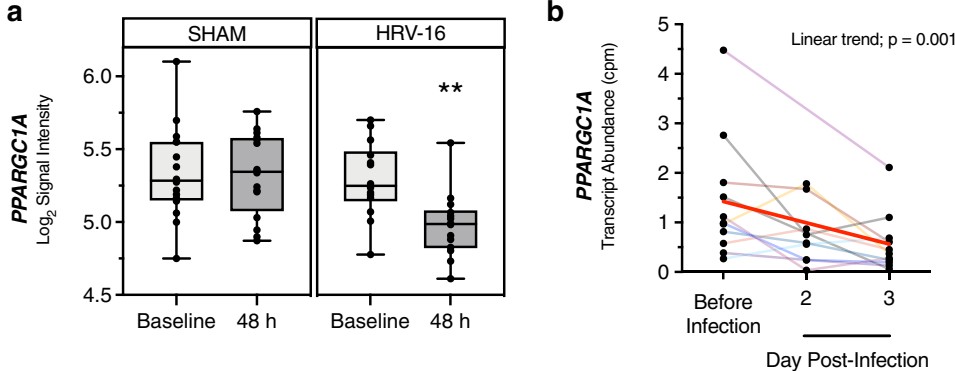

**Fig. 8 PGC-1α is a therapeutic target in clinical HRV infections. a** Healthy volunteers were infected with sham (saline) or HRV-16 for 48 h and nasal scrapings were obtained prior to infection (baseline) and 48 h post infection for microarray analysis. Normalized *PPARGC1A* transcripts were only downregulated in HRV-16 patients. Data were analyzed using Transcriptome Analysis Console (TAC) software (Affymetrix) by eBayes ANOVA, sham: $p = 0.88$ ($n = 16$ patients), HRV-16 infected **$p = 0.0053$ ($n = 14$ patients). **b** *PPARGC1A* RNAseq transcript abundance from nasal scrapings obtained during an HRV-39 clinical infection study in healthy volunteers. Data are represented as individual patient lines. Data were fit to a linear model and the coefficient of the linear trend was tested (determining that the slope of the line with time is nonzero). Linear trend $p = 0.0014$, $n = 11$ patients.

PneumaCult-EX Basal Medium supplemented with 50× Supplement, hydro-cortisone, fluconazole, and penicillin/streptomycin (PneumaCult-EX) for 72 h. At 90% confluence, HBE cells were lifted using TrypLE Select and seeded into bovine collagen-coated transwells (1.12 cm$^2$, 0.4 μm pore size) at a density of $2.0 \times 10^5$ cells per insert in PneumaCult-EX. PneumaCult-EX was also added to the basolateral compartment, and HBE cells were maintained at 37 °C, 5% $CO_2$. At 48 h, apical media were removed to expose the HBE cells to air, and basolateral PneumaCult-EX was replaced with PneumaCult-ALI Basal Medium containing 10× Supplement, fluconazole, and penicillin/streptomycin (PneumaCult-ALI Differentiation Medium). Cells were fed basolaterally every 48 h with PneumaCult-ALI Differentiation Medium freshly supplemented with 100× Supplement, hydrocortisone, and heparin prior to each feeding. At 14 days post-transwell seeding, ALI cultures were washed apically once per week with Dulbecco's phosphate-buffered saline (DPBS) to remove excess mucus as a result of secretory cell differentiation. Highly differentiated ALI cultures were used for all experiments 5 weeks post-transwell seeding.

**HRV inoculation**. ALI cultures were rinsed apically with DPBS to remove excess mucus and fed basolaterally with PneumaCult Basal Media (no supplements) and allowed to rest for 2 h at 37 °C, 5% $CO_2$. ALI cultures were infected apically with 100 μL of HRV-C15, HRV-1A, or HRV-16 ($10^9$ RNA copy number) diluted in 25 mM HEPES in F12 or mock infected with 100 μL 25 mM HEPES in F12 and maintained at 34 °C, 5% $CO_2$. At 2 h post infection, ALI cultures were washed twice with DPBS to remove non-internalized HRV-C15 or mock solution and maintained at 37 °C, 5% $CO_2$ until the experimental endpoint. For HRV replication kinetic experiments quantifying HRV-C15 intracellular replication and apical release over time, at 2 h post infection ALI cultures were washed apically five times with DPBS to remove virtually all non-internalized HRV-C15 or mock solution and maintained at 37 °C, 5% $CO_2$. For experiments exceeding 24 h, cells were fed basolaterally with PneumaCult-ALI Basal Media every 24 h to maintain viability.

**HRV propagation**. The pC15-RZ plasmid containing a full-length HRV-C15 sequence was used to produce the HRV-C15 RNA genome. The plasmid was linearized using the restriction enzyme *Bst*BI and subjected to T7-dependent in vitro transcription using the RiboMAX Large Scale RNA Production System. DNase treatment was used to degrade the DNA template and the viral RNA quality was validated using 0.8% agarose gel electrophoresis. The HRV-C15 RNA genome was transfected using Lipofectamine RNAiMax into WI-38 cells grown to confluence in T175 cm$^2$ flasks using Dulbecco's modified Eagle's medium (DMEM) supplemented with 10% FBS, 2 mM L-glutamine, 1% non-essential amino acids, and 1% sodium pyruvate. HRV-C15 was allowed to replicate in WI-38 cells until cell rounding was observed. HRV-16 and HRV-1A were originally purchased from ATCC and were propagated in WI-38 fibroblasts[51] or H1-HeLa cells[52] according to our established protocols.

**Viral purification and dialysis**. Lysed infected WI-38 supernatants (HRV-C15 and HRV-16) or H1-HeLa supernatants (HRV-1A) were incubated with RNase A for 30 min at 34 °C to remove free single-stranded RNA. N-laurosarcosine and 2-mercaptoethanol were added to dissociate the virus from the epithelial membranes and to reduce disulfide bonds and degrade residual cellular components, respectively. Supernatants were underlaid with a sucrose solution (30% sucrose, 20 mM Tris acetate, 0.1 M NaCl) and centrifuged at 105,000 × g for 5 h at 16 °C using an SW28 rotor in a Beckman Optima XL-100K Ultracentrifuge. Centrifugation yielded

HRV-containing sucrose. This was combined 1:1 with 50 mM HEPES in F12, filter sterilized through a syringe filter (0.45 μm) and dialyzed (20,000 MW) against 25 mM HEPES in F12 media for 18 h at 4 °C to remove sucrose. HRV was maintained in aliquots at −80 °C for long-term storage.

**Pharmacological interventions**. Prior to experiments using pharmacological interventions (oligomycin A, 2-DG, ZLN005, SR18292, and MitoTEMPO), ALI cultures were washed apically once in DPBS to remove excess mucus, transitioned to PneumaCult-ALI Basal Medium (no supplements or antibiotics), and allowed to rest for 2 h at 37 °C, 5% $CO_2$ prior to treatments. ALI cultures were first infected with HRV or mock solution for 2 h at 34 °C, 5% $CO_2$. Non-internalized HRV or mock solution was removed with one DPBS wash, and pharmacological reagents were added apically in 25 mM HEPES in F12 and basolaterally in PneumaCult-ALI Basal Medium and maintained at 37 °C, 5% $CO_2$. For experiments exceeding 12 h, apical treatments were removed at 12 h post exposure and inhibitors were refreshed once basolaterally and maintained until the experimental endpoint.

**Immunofluorescence**. ALI cultures were fixed in 100% ice-cold methanol (target antigens: occludin, ZO-1, and dsRNA) for 15 min at 4 °C, or 4% paraformaldehyde (target antigen: β-tubulin) for 30 min at room temperature (RT). Cells were rinsed twice in 1× PBS and blocked and permeabilized in 10% bovine serum albumin (BSA) in 0.1% Triton X-100 in 1× PBS for 2 h at RT. ALI cultures were incubated with primary antibodies: ZO-1 (1:200), occludin (1:200), dsRNA (1:200), and β-tubulin (1:500), all diluted in 2% BSA in 0.1% Triton X-100 in 1× PBS overnight at 4 °C. ALI cultures were washed three times in 0.1% Triton X-100 in 1× PBS for 5 min and incubated with appropriate conjugated fluorescent secondary antibodies: AlexaFluor488 goat anti-mouse (1:200), AlexaFluor488 goat anti-rabbit (1:200), and AlexaFluor647 goat anti-mouse (1:200) for 2 h at RT. ALI cultures were rinsed three times in 0.1% Triton X-100 in 1× PBS for 5 min, counterstained with 4′,6-diamidine-2′-phenylindole dihydrochloride at 1:50,000 (0.2 μg/mL) for 30 min at RT and mounted onto Superfrost Plus slides with FluorSave Reagent. Confocal imaging was performed using a Leica TCS SP8 resonance scanning confocal microscope (×25 water immersion objective) using Leica LAS X software (v.3.5.5.19976).

**Immunohistochemistry**. ALI cultures were fixed in 10% neutral-buffered formalin, embedded in paraffin wax, and sectioned to 4 μm thickness onto Superfrost Plus slides. Immunohistochemistry highlighting ciliated cells (β-tubulin) was performed by de-paraffinizing in two changes of xylene and rehydrated through graded ethanol solutions (100%, 95%, and 70% EtOH) and distilled $H_2O$. Endogenous peroxidase was quenched with 3% $H_2O_2$ for 20 min and sections were blocked in 2% horse serum for 40 min at RT. Avidin and biotin blocks were added for 15 min each, rinsed in PBS, and incubated with mouse anti-β-tubulin (1:500) in 2% horse serum for 90 min at RT. Sections were incubated with biotinylated horse anti-mouse antibody (1:200) in 2% horse serum for 40 min, washed three times for 5 min in PBS, and incubated with avidin–biotin complex for 30 min. Sections were rinsed in PBS and $H_2O$ before applying 3,3′-diaminobenzidine peroxidase substrate for 20 s. Sections were counterstained in Gills II hematoxylin for 5 min and rinsed in warm tap water for 5 min. Slides were dehydrated through reverse graded ethanol solutions (70%, 95%, and 100% EtOH) and cleared in two changes of xylene before mounting with Permount. Brightfield microscopy on ALI culture histological sections was performed using a ×100 objective on an EVOS XL Cell Imaging System.

**Immunoblotting**. ALI cultures were collected in lysis buffer (1% Triton X-100 in 1× MES buffered saline pH 7.4, containing anti-protease tablets, 2 mM sodium orthovanadate, 20 mM sodium pyrophosphate, 50 mM sodium fluoride, and 1 mM phenylmethylsulfonyl fluoride), sonicated for 30 min, and centrifuged at $10,000 \times g$ for 10 min to collect soluble protein. Protein was quantified using a BioRad DC Protein Assay. Whole-cell lysates were separated by 10% sodium dodecyl sulfate-polyacrylamide gel electrophoresis (SDS-PAGE) and transferred to a 0.45 μm nitrocellulose membrane, blocked in 0.1% TBST containing 5% BSA for 1 h at RT, incubated overnight with rabbit anti-PGC-1α (1:1000) in 5% BSA in 0.1% TBST at 4 °C. Membranes were washed three times for 5 min with 0.1% TBST and incubated with anti-rabbit IgG horseradish peroxidase (HRP)-linked antibody (1:10,000) in 5% BSA in 0.1% TBST for 1 h prior to detection using Pierce™ ECL substrate. Housekeeping protein GAPDH was probed with mouse anti-GAPDH (1:40,000) in 5% BSA in 0.1% TBST and detected with anti-mouse IgG HRP-linked antibody (1:10,000) in 5% BSA in 0.1% TBST using the same protocol.

**RNA purification and real-time qPCR**. Intracellular RNA was isolated using the column-based Nucleospin RNA Kit and eluted in RNase/DNase free water. RNA purity and concentration were quantified by a NanoDrop 2000 spectrophotometer. HRV-C15 RNA from apical wash samples were isolated using the QIAamp Viral RNA Mini Kit. Intracellular HRV-C15 and apical wash HRV-C15 were subjected to RT-qPCR using primers and TaqMan probe (FAM/MGB) directed to the 5′-untranslated regions of HRV, such that the primers and probe can detect all strains of HRV used in this study. Primer and TaqMan probe (FAM/MGB) sequences were designed in-house to *IFNL1*, *IFNB1*, and *RSAD2*. Specific TaqMan Gene Expression Assays were obtained for *SLC2A1*, *TFAM*, *HK1*, *HK2*, *PPARGC1A*, and *POLG*. All RT-PCR was performed using FastStart Universal Probe Master (ROX) in a 7500 Fast Real-Time PCR System (Applied Biosystems). Samples were quantified using the $2^{-\Delta\Delta CT}$ method[53], and standardized to the housekeeping gene, *GAPDH*. Primer and TaqMan probe sequences are provided in Supplementary Table 2.

**FITC-dextran permeability**. Paracellular permeability was measured by apical to basolateral movement of FITC-dextran (3–5 kDa). The ALI cultures were infected according to the standard HRV infection protocol with or without pharmacological interventions. At each time point, the apical compartment was filled with 200 μL (2 mg/mL) FITC-dextran diluted in 25 mM HEPES in F12, and the basolateral compartment was filled with fresh PneumaCult-ALI Basal Medium. If pharmacological interventions were used, they were maintained throughout the duration of the assay diluted in the apical FITC-dextran solution and the basolateral PneumaCult-ALI Basal Medium. The ALI cultures were incubated with FITC-dextran for 2 h at 37 °C, 5% $CO_2$. Input FITC-dextran (2 mg/mL) from each individual experiment was serially diluted in PneumaCult-ALI Basal Medium to create a standard curve for absorbance quantification. ALI culture basolateral media samples and FITC-dextran standard curve were plated into black 96-well microtiter plates and read on a PerkinElmer fluorescence plate reader.

**Transepithelial electrical resistance**. TEER was obtained as one quantitative measurement of paracellular permeability by using a dual planar chopstick electrode EVOM Voltohmmeter. ALI cultures were rinsed apically and basolaterally with warm DPBS supplemented with $Ca^{2+}$ and $Mg^{2+}$ to remove secreted mucus and media residue prior to measurement. Chopstick electrodes were submerged in 0.5 mL DPBS apically and 1.5 mL DPBS basolaterally to record TEER in ohms (Ω). Resistances were recorded as $\Omega \times cm^2$ and calculated as percent change from noninfected ALI cultures at each time point.

**Matched proteomic and metabolomic experiments**. ALI cultures derived from six lung donors were infected with HRV-C15 or mock diluted in 25 mM HEPES in F12 for 2 h, as described above. For metabolomics, each plate of cells contained an internal control of cell-free media sampled at each time point to account for the natural rate of media degradation at 37 °C, 5% $CO_2$. At each time point (4, 12, and 24 h), ALI basolateral media were collected and incubated 1:1 in ice-cold 100% methanol for 30 min on ice, vortexing every 10 min. Basolateral media were centrifuged at $20,000 \times g$ for 10 min at 4 °C, and further diluted in 50% methanol prior to mass spectrometry plate loading. All donor samples were collected, processed, and run on the same mass spectrometry plate to minimize machine analysis variability. For matched proteomic analysis, the remaining ALI insert was lysed in 1% SDS, 100 mM ammonium bicarbonate, 1 mM EDTA, and 1× protease inhibitors. Protein was briefly sonicated on ice, centrifuged at $10,000 \times g$, and stored at −80 °C until TMTsixplex isobaric labeling.

**Metabolite analysis**. General metabolomics runs were performed on a Q Exactive™ HF Hybrid Quadrupole-Orbitrap™ Mass Spectrometer coupled to a Vanquish™ UHPLC System. Chromatographic separation was achieved on a Syncronis HILIC UHPLC column (2.1 mm × 100 mm × 1.7 μm) using a binary solvent system at a flow rate of 600 μL/min. Solvent A, 20 mM ammonium formate pH 3.0 in mass spectrometry grade $H_2O$; Solvent B, mass spectrometry grade acetonitrile with 0.1% formic acid (% v/v). The following gradients were used: 100% Solvent B (0–2 min), 100–80% Solvent B (2–7 min), 80–5% Solvent B (7–10 min), 5% Solvent B (10–12 min), 5–100% Solvent B (12–13 min), and 100% Solvent B (13–15 min). A sample injection volume of 2 μL was used. The mass spectrometer was run in negative full scan mode at a resolution of 240,000 scanning from 50 to 750 *m/z*. Metabolite analyses were completed using El Maven (v.0.12.0), a mass spectrometry data analysis software package[54,55]. Metabolites were identified by matching observed *m/z* signals (±10 p.p.m.) and chromatographic retention times to those observed from the reference Mass Spectrometry Metabolite Library.

**Metabolomic analysis**. Metabolomic analyses and statistical figures were generated using MetaboAnalyst 4.0. For the comprehensive heatmap of all donors and timepoints (Fig. 3a), the data were normalized to the pooled noninfected donors at 4 h. For individual time point heat maps (Fig. 3b and Supplementary Fig. 2a, d), data were normalized to matched timepoint noninfected donors. The samples were scaled (mean-centered and divided by the standard deviation of each variable). We applied Euclidean distance measure and Ward clustering algorithm without sample reorganization to cluster group samples. Quantitative enrichment analysis was performed by discrete (classification) group labeling and compound name or compound ID type. Samples were normalized from matched timepoint noninfected donors, scaled, and the Small Molecule Pathway Database metabolite set library was used.

**TMTsixplex™ isobaric labeling for protein analysis**. Protein concentrations were determined by a NanoDrop 2000 spectrophotometer at 280 nm. One hundred micrograms of protein was combined with lysis buffer (1% SDS, 100 mM ammonium bicarbonate, 1 mM EDTA, and 1× protease inhibitors) to a final volume of 100 μL. Samples were reduced in 200 mM tris(2-carboxyethyl)phosphine, for 1 h at 55 °C, and reduced cysteines were alkylated by incubation with iodoacetamide solution (50 mM) for 30 min at RT. Samples were precipitated by acetone/methanol, and 600 μL ice-cold acetone was added followed by incubation at −20 °C overnight. A protein pellet was obtained by centrifugation ($8000 \times g$, 10 min, 4 °C) followed by acetone drying (2–3 min). The precipitated pellet was resuspended in 100 μL of 50 mM triethylammonium bicarbonate buffer followed by trypsin digestion (5 μg trypsin per 100 μg of protein) overnight at 37 °C. TMTsixplex™ Isobaric Labeling Reagents were resuspended in anhydrous acetonitrile and added to each sample (41 μL TMT reagent per 100 μL sample) and incubated at RT for 1 h. The TMT labeling reaction was quenched by 2.5% hydroxylamine for 15 min at RT. TMT-labeled samples were combined and acidified in 100% trifluoroacetic acid to pH < 3.0 and subjected to C18 chromatography (Sep-Pak) according to the manufacturer's recommendations. Samples were stored at −80 °C before lyophilization, followed by resuspension in 1% formic acid before liquid chromatography and tandem mass spectrometry (LC-MS/MS) analysis.

**Liquid chromatography and tandem mass spectrometry**. Tryptic peptides were analyzed on an Orbitrap Fusion™ Lumos™ Tribrid™ Mass Spectrometer operated with Xcalibur (version 4.0.21.10) and coupled to an EASY-nLC™ 1200 System. Tryptic peptides (2 μg) were loaded onto a C18 trap column (75 μm × 2 cm) at a flow rate of 2 μL/min of Solvent A (0.1% formic acid in LC-MS grade $H_2O$). Peptides were eluted using a 120 min gradient from 05 to 40% (5–28% in 105 min followed by an increase to 40% Solvent B in 15 min) of Solvent B (0.1% formic acid in 80% LC-MS grade acetonitrile) at a flow rate of 0.03 μL/min and separated using a C18 analytical column. Peptides were electrosprayed using 2.1 kV voltage into the ion transfer tube (300 °C) of the Orbitrap Lumos operating in a positive mode. The Orbitrap first performed a full MS scan at a resolution of 120,000 full-width at half-maximum to detect the precursor ion having an *m/z* between 375 and 1575 and a +2 to +4 charge. The Orbitrap AGC (Auto Gain Control) and the maximum injection time were set at $4 \times 10^5$ and 50 ms, respectively. The Orbitrap was operated using the top speed mode with a 3 s cycle time for precursor selection. The most intense precursor ions presenting a peptidic isotopic profile and having an intensity threshold of at least 2e4 were isolated using the quadrupole (isolation window (*m/z*) of 0.7) and fragmented using higher-energy collisional dissociation (38% collision energy) in the ion routing multipole. The fragment ions ($MS^2$) were analyzed in the Orbitrap at a resolution of 15,000. The AGC and the maximum injection time were set at $1 \times 10^5$ and 105 ms, respectively. The first mass for the MS2 was set at 100 to acquire the TMT reporter ions. Dynamic exclusion was enabled for 45 s to avoid the acquisition of the same precursor ion having a similar *m/z* (±10 p.p.m.).

**Proteomic analysis**. Spectral data were matched to peptide sequences in the human UniProt protein database using the Andromeda[56] algorithm as implemented in the MaxQuant[57] software package (v.1.6.0.1), at a peptide-spectrum match false discovery rate of <0.01. Search parameters included a mass tolerance of 20 p.p.m. for the parent ion, 0.5 Da for the fragment ion, carbamidomethylation of cysteine residues (+57.021464 Da), variable N-terminal modification by acetylation (+42.010565 Da), and variable methionine oxidation (+15.994915 Da). TMT6-plex labels 126 to 131 were defined as labels for relative quantification. The cleavage site specificity was set to Trypsin/P (search for free N terminus and for only lysines), with up to two missed cleavages allowed. Significant outlier cutoff values were determined after log(2) transformation by boxplot-and-whiskers analysis using the BoxPlotR tool[58]. The data were deposited into ProteomeXchange via the

PRIDE database (PXD024591), and analysis was performed using Metascape[59]. We selected and uploaded multiple gene lists containing all statistically enriched terms that were identified; accumulative hypergeometric *p* values and enrichment factors were calculated and used for filtering. The remaining significant terms were then hierarchically clustered into a tree based on kappa-statistical similarities among their gene memberships. Then 0.3 kappa score was applied as the threshold to cast the tree into term clusters. A subset of representative terms was selected from the full cluster and converted them into a network layout. All PPIs among each input gene list were extracted from PPI data source and formed a PPI network. GO enrichment analysis was applied to the network to assign biological meanings.

**Integrated metabolomic and proteomic analysis**. Integrated pathway analysis was performed using MetaboAnalyst 5.0. Gene lists and compound lists were entered as fold change relative to matched timepoint noninfected cells. Official gene symbol was selected for gene list ID type and compound name selected for compound ID type. The integration occurred in the universe defined by metabolic pathways, which includes pathways containing both metabolites and metabolic genes. Enrichment analysis was performed by hypergeometric test, and topology measured degree centrality. We used combined queries as the integration method.

**Oxygen consumption quantification**. Mitochondrial respiration was measured in live ALI cultures. For HRV-C15 baseline oxygen consumption studies at 4, 8, and 12 h post infection, a reverse infection time course was performed in which ALI cultures were infected every 4 h until 12 h to perform the measurements simultaneously to minimize timepoint and plate variability. In all other instances, ALI cultures were infected with HRV-C15 simultaneously and analyzed by Seahorse in the same plate. ALI cultures were infected according to the standard infection protocol for 4, 8, and 12 h in PneumaCult Basal Medium and excised from transwells using 2 mm biopsy punches to produce three punches per insert per condition. Biopsy punches were carefully loaded into Seahorse XFe24 microplates, and 0.5 mL/well Seahorse XF DMEM Medium supplemented with 4 mM L-glutamine, 24 mM glucose, and 2 mM sodium pyruvate and adjusted to pH 7.4. ALI biopsy punches were acclimated for 1 h at 37 °C without $CO_2$ and transferred to a Seahorse XFe24 Analyzer for analysis. All OCR and ECAR measurements were performed three times using a 3–2–3-min mix–wait–measure cycle.

**ROS quantification**. ROS confocal imaging was performed on live ALI cultures. ALI cultures were infected with HRV-C15 or mock, as described above. At each time point (4, 8, and 12 h), cells were incubated apically with 5 μM CellROX Green Reagent and 5 μg/mL Hoechst 33342 diluted in 25 mM HEPES in F12 for 30 min at 37 °C, 5% $CO_2$. CellROX and Hoechst 33342 were removed by rinsing ALI cultures three times in DPBS. Live cell imaging of oxidative stress (ROS) was quickly performed on a Leica TCS SP8 resonance scanning confocal microscope (×25 water immersion objective) using Leica LAS X software (v.3.5.5.19976). Three fields of view per ALI insert were captured and relative fluorescence intensities were quantified using the ImageJ software (version 1.52i, National Institutes of Health, USA).

**Cell viability**. Two different methods of necrosis assessment were performed on the ALI cultures. Lactate dehydrogenase (LDH) was quantified by measuring the apical and basolateral release of LDH using a colorimetric CyTox96® Non-Radioactive Cytotoxicity Assay according to the manufacturer's recommendations. LDH standard curves were generated for each donor by serial dilution of one total LDH cell lysate. Live cell propidium iodide (PI) staining was performed to assess in situ loss of membrane integrity. ALI cultures were incubated with 5 μg/mL Hoechst 33342 in 25 mM HEPES in F12 media added apically, for 10 min at 37 °C, 5% $CO_2$. Hoechst was removed with two rinses in PBS. One ALI insert at a time, 200 μg/mL PI in 25 mM HEPES in F12 was added apically for 5 min at 37 °C, 5% $CO_2$. ALI insert was quickly excised from the transwell using a No. 21 surgical blade, placed onto a microscope slide, and imaged by tile scan on a Leica TCS SP8 resonant scanning confocal microscope (×25 water immersion objective) with the Leica LAS X software (v.3.5.5.19976). Image analyses were performed using the ImageJ software (version 1.52i, National Institutes of Health, USA).

**Experimental HRV infection studies**. Expression of *PPARGC1A* was examined from two experimental HRV infection studies in normal volunteers. The first was a previously published study in which subjects with no detectable neutralizing antibodies were infected with HRV-16[8]. In this study, RNA was isolated from nasal epithelial scrapings taken before infection (baseline) and at 48 h after infection, the latter corresponding to symptomatic illness. Samples were processed and gene expression of >47,000 transcripts was analyzed using Affymetrix gene array chips. The full dataset was submitted to the GEO (GSE11348) and the expression of *PPARGC1A* was examined using this dataset. Robust multiarray averaging, quantile normalization, and median polishing on logged probe set intensity values were performed in Transcriptome Analysis Console (TAC) software v4.0 (Affymetrix). The TAC software was then used to perform the eBayes analysis of variance

(ANOVA) method, as is appropriate for microarray data to produce descriptive statistics for data categorization[60]. To confirm the 2008 study data, we also analyzed the expression of *PPARGC1A* from a second experimental HRV infection study that is currently nearing completion in our laboratory. In this study, both normal control subjects and smokers with normal lung function and no preexisting neutralizing antibodies were experimentally infected with GMP grade HRV-39, and RNA was isolated from nasal scrapings taken before infection and at 48 and 72 h post infection and were processed for gene expression analysis using RNAseq. To compare both to our current in vitro data and to the prior published in vivo gene array study described above, we analyzed the RNAseq dataset from normal subjects only ($n = 11$) for expression of *PPARGC1A*. All subjects who participated in this study gave informed consent and the protocol was approved both by the Conjoint Health Research Ethics Board of the University of Calgary (Ethics ID: REB15-0991) and by Health Canada.

**Statistical analysis**. The normality of datasets was tested using the Kolmogorov–Smirnoff test. Data that were normally distributed were analyzed by one-way ANOVA with appropriate post hoc tests. Paired data were compared using two-tailed paired Student's *t* test. Nonparametric data were analyzed using Kruskal–Wallis ANOVA with Dunn's post hoc analysis. Significance was assumed for values of $p < 0.05$.

**Reporting summary**. Further information on research design is available in the Nature Research Reporting Summary linked to this article.

## Data availability

Proteomics RAW data were deposited to ProteomeXchange via the Proteomics Identification Database (PRIDE) under accession number PXD02459. Metabolite LC-MS data were deposited to Metabolomics Workbench under accession number ST001774. Clinical HRV infection study data were obtained from the National Center for Biotechnology Information/Gene Expression Omnibus accession number GSE11348. Source data are provided with this paper.

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

## Acknowledgements

We thank Dr. James Gern and Dr. Yury Bochkov at the University of Wisconsin for generously providing the pC15-RZ plasmid containing a full-length HRV-C15 sequence. We also thank Dr. Ian Lewis and Dr. Marija Drikić for metabolomics data acquisition and processing at the Calgary Metabolomics Research Facility (CMRF), which is supported by the International Microbiome Center and the Canada Foundation for Innovation (CFI-JELF 34986). Finally, we thank Daniel Young and the Southern Alberta Mass Spectrometry (SAMS) core facility for proteomic analysis, Dr. Jane Shearer for Seahorse Analyzer access, and Dr. Mahmoud Mostafa and Dr. Thomas Mahood for bioinformatics guidance. This study was supported by the Canadian Institutes of Health Research (CIHR) (grant number PJT-159635) and the Natural Science and Engineering Research Council of Canada (NSERC) (grant number RGPIN-2018-03861). D.P. holds a Tier 1 Canada Research Chair in Inflammatory Airway Diseases. A.N.M. acknowledges funding from Asthma Canada, the Canadian Allergy, Asthma and Immunology Foundation (CAAIF), and The Lung Association, Alberta & NWT.

## Author contributions

Conceptualization: A.N.M.; methodology: A.N.M.; validation: A.N.M.; formal analysis: A. N.M., F.L., and A.D.; investigation: A.N.M.; resources: B.G.Y., A.D., and D.P.; writing—original draft: A.N.M.; writing—review and editing: A.N.M., F.L., B.G.Y., A.D., and D.P.; visualization: A.N.M. and F.L.; supervision: D.P.; funding acquisition: D.P.

## Competing interests

The authors declare no competing interests.
