## [Peer Review File · Nature Communications]

REVIEWER COMMENTS

Reviewer #1 (Remarks to the Author):

I appreciate the opportunity to review this interesting manuscript. As my expertise is metabolomics and proteomics analysis (as well as its integration) in respiratory infection, my review focuses on these areas. As for the validity of experimental models, I humbly defer to the other reviewers. In this study, the authors utilized experimental rhinovirus infection in highly differentiated human bronchial epithelial cells grown at ALI. The authors found that 1) metabolism regulates airway epithelial barrier function, 2) RV-C15 impairs early barrier function prior to viral pattern recognition receptor driven responses through metabolic mechanisms, and 3) PGC-1 α plays a role in regulating barrier function. The authors speculated that PGC-1 α may be a novel therapeutic target for recovering RV-induced barrier loss while decreasing viral titers.

GENERAL COMMENTS

I highly applaud the authors' approaches to use well-thought experimental designs and applications of state of the art metabolomics and proteomics approaches. However, this study has several limitations, such as 1) potential lack of power in these analyses, 2) multiple hypothesis testing, 3) lack of clarity in the statistical analysis, and 4) opportunity to integrate different omics information. Below, I included several comments that might help the authors improve the clarity of the manuscript.

SPECIFIC COMMENTS

ABSTRACT:

Page 2, para 1: "opportunistic co-infections". The reviewer is unsure whether RV is a common agent for opportunistic coinfections even in immunocompromised patients. Please clarify or revise this one.

INTRODUCTION:

Page 3, para 1: "Although it has been clearly established...". Please add citation(s) that supports this notion.

Page 3, para 3: "We employed an unbiased ... proteomics and metabolomics". The term "unbiased", while often used, is not appropriate but misleading. "Unbiased" indicates that the point estimate equals to the true value, which is not guaranteed by these testing. Please replace it with a more appropriate word, such as "untargeted".

METHODS:

Pages 38-39: The validity of these metabolomics and proteomics measurements is unclear. This is very important information to assess the validity of the study. The authors should report the within-run and between-run CV or the median relative standard deviation for the internal standards.

Furthermore, the authors should report the number of metabolites and proteins measured / identified. What are the reference libraries to identify / name these metabolites and proteins? Currently, it is difficult to assess the validity of these testing.

Page 44: The analytical section consists only of four lines, which is highly insufficient. The reviewer has several questions in order to assess the validity of the analysis, such as:

- How was the multiple testing problem addressed, in the setting of sample size of 6?
- Which software and packages were used (e.g., MSEA with MetaboAnalyst for the pathway analysis)?
- How were the within-individual/sample correlations addressed?
- How were the pathway analyses performed? What were the reference libraries?

All of the information is essential to guarantee the reproducibility.

RESULTS

Page 5, para 3: This is important. The sample size for the metabolomics and proteomics testing was only 6 while many individual metabolites/proteins as well as pathways were tested (the reviewer speculate that hundreds of hypotheses * multiple time points were tested). This will certainly lead to type 1 errors. How did the author address this critical issue? Please clarify.

Page 6, para 1: Based on the proteomic enrichment analysis, the "metabolomic process" is the most significant. However, this is a very generic "process" which consists of numerous "processes". Can the author comment on this?

Additionally, most analyses were performed as a series of cross-sectional analysis. There is an opportunity to examine the longitudinal change in these proteins and metabolites over time after experimental infections.

It is intriguing to see that significant lipid metabolites are substrates of cell membranes (e.g., ethanolamines, L-serine [a substrate of sphingolipids/ceramides]). It may suggest that these metabolites are also results of cell destruction/death instead of increased fatty acid synthesis. The reviewer would like to hear how the authors interpret this important finding.

Pages 6-7: While the sample size may be limited, the study has interesting data – longitudinal measurements of the metabolome, proteome, and six mRNAs. Currently, these data are examined individually and cross-sectionally. There is an interesting opportunity to integrate these data to derive further knowledge. The authors may want to consider additional analyses. There are many techniques/approaches available. Here is a good review article that the might be helpful – Noell et al. Eur Respir Med 2018.

Reviewer #2 (Remarks to the Author):

COMMENTS TO THE AUTHORS:

Common cold infections, caused by human rhinovirus (HRV) are directly associated with exacerbations in those typically with lower airway diseases, including asthma, COPD and CF. Development of vaccines is improbable for these viruses since there are over 150 serotypes known to date and more being identified. As a result, alternative strategies are being sought to combat viral infection. In this study, Michi and colleagues assessed the effects of HRV infection on airway epithelial cell metabolism and associated barrier function. There is very sound premise and execution of this study and the authors should be complemented for such a comprehensive study. Despite this, there are numerous shortfalls that render the manuscript in its current form unsuitable for publication. Firstly, the authors have concentrated on HRV-C which predominantly is associated with exacerbation in young children, however primary cells were derived from adults (only 1- adolescent (13yrs). Secondly, cells were derived from a presumed healthy population whereas the target for the proposed therapies are those with chronic airway diseases whose baselines may be different to those assessed here. Furthermore, the used of uninfected mock controls may not have been the most appropriate, as the effect viral binding were not thoroughly assessed. Study limitations were not appropriately acknowledged and discussed and there were also several methodological questions that leave the reader unclear about how to reproduce the study.

MAJOR ISSUES:

1. The premise of this paper appears to explore the potential of targeting cellular metabolism and its associated affects on airway epithelial barrier function in response to HRV infection. The rationale behind this is not queried as it has good merit, however, since the target was not age specific there is insufficient clarity as to why the majority of the study solely investigated host effects to HRV-C and not HRV A or B. (as initially perform in the study). This further compounded by the fact that the cohorts used to obtain primary airway cells were all older adults, whereas HRV-C causes pronounced exacerbations in young children. There is evidence that no HRV serotype predominates in adults (McCulloch et al. 2014. Am J Clin Path 142:165) and conflict needs to be addressed.
2. The authors have utilized a specialized medium to grow and induced airway differentiation

(PneumCult). However, there are studies suggesting various levels of differentiation obtained with different commercial products (PMID: 32332878, PMID: 32967385) and infer differentiation of primary cells into more the nasal phenotype with this product. The authors will need to justify their differentiation into lower airway cells more strongly in the context of this.

3. It was difficult to know where appropriate mock controls were used in this study. Since the authors argue, that effects are the result of active infection (of which the experiments were conducted appropriately) the necessary mock infection control to this reviewer would have been UV- inactivated virus. Although some of the generated data does illustrate the inclusion of this type of mock control, the methods suggestion the inclusion of a 'no virus' volume (HEPES based). The reviewer then interpreted the rest of the data to have include this non-viral control. The authors will need to address and interpret their data then in this context, as it's a limitation.

4. The authors identified transcriptional and translational modifications at the same time point (4 h), which is a very unique finding. Usually post transcriptional modifications occur sometime after transcriptional changes. The authors have not addressed this this observation, just the speed at which observations are made.

5. The authors will need to attempt some explanation to the clustering of their cohort (Fig 2B right hand panel). Appears 4 participants clustered and 2 did not. Was this age, gender corrected? Cause of death skewed? This information is needed.

6. Fig. 3D should be expanded to include a 24 hr time point so as to be definitive in the return to basal state with Oligomycin A treatment.

7. The inclusion of experimental infection data into this manuscript is one of the major strengths to this study. Although these prior performed studies were conducted by the same group, the way the manuscript was written infers they were specifically performed as part of this study. The first study was published in 2008 and the second comes from a second as yet unpublished study. The methods section, results, abstract and discussion need to be modified that you fortuitously were able to conduct analyse on these other studies. If the second study is as yet unpublished and the data to be included in this study, the cohort and information needs to become part of this study.

8. Furthermore, these other corroboration populations samples only nasal epithelium. Although results mirror those found from lower airway epithelial cells which is a strength to the study, there are implications considering the upper and lower airway certainly have very different functions, especially when it comes to pathogen exposure and host responses. The authors will need to address this.

9. The discussion lack any comments around those that need the therapeutics the most, ie those with COPD, asthma and CF. There needs to be discussion around the disease cohorts as well as age issues.

MINOR ISSUES:

1. Introduction page 2-3, second paragraphs, 'over 150 serotypes... not species' also 'three species not genetic groups'

2. Page 4 Results "Barrier function disruption was dependent on actively replicating virus, as neither replication deficient UV-treated HRV-C15." Sentence appears incomplete.

3. Page 6, Figure 2A left... should be 2B left.

4. Page 33: Please state the sources of HRV-1A and 16.

5. Page 33 Please include TEER measurement methodology in appropriate sections. Was this used to corroborate ALI? Add required information in the appropriate sections.

6. Page 37. Immunoblotting sentence. "Membrane was washed"..... replace with 'Membranes were washed...'

7. Page 37. Probes for SLC2A1 missing (as included in graph 3A) from methods section.

8. Page 38. Include the FITC-Dextran size in methods section (kDa).

9. Page 41. 3 punch biopsies from the one insert are not considered technical replicates. Inserts will differ in their differentiation and their infectivity and repeated inserts are required for these measurements (oxygen consumption quantification).

10. Call confocal images will need to be addressed. It appears that they were counterstained with DAPI but it is extremely difficult to visualize this in many panels. All panels need to be normalized for both TJ and DAPI.

11. Figure 1E . Non-infected dsRNA panel scale (25um) not equivalent to rest of panel (50um). Please replace with appropriate panel.

12. Supplement Figure 1D UVHRVC15 panel again different scale. Replace.

Reviewer #3 (Remarks to the Author):

In the current study, the authors look at the impact of RhinovirusC15 on barrier function in ALI models. They demonstrate that infection induces a metabolic remodelling that contributes to the disintegration of barrier function. Pharmacological intervention in this process can reduce viral replication.

Major

1. The generalisation of this finding – is it restricted to just one Rhinovirus isolate? Could you confirm in another C virus? But is it more broad, many viruses disrupt barrier function, is it unique to rhinovirus?
2. Many of the introductory statements leading into a block of work could be cut as they feel either repetitive (referencing early parts of the manuscript) or over-interpretive e.g. 'Given that changes in barrier function were not dependent upon conventional host innate antiviral signaling' on p5. This statement is not supported, it is inference from the timing of response.
3. Much of the text/ discussion emphasises the novelty/ first that this work has been done – this is unnecessary and potentially inaccurate. For example Unger et al examined ROS on barrier function in RV infection (with a different strain of RV: <https://pubmed.ncbi.nlm.nih.gov/24429360/>). Would be clearer to state the results and let the reader draw their own interpretation of the impact/ importance.
4. More clarity on the use of human challenge needs to be made in the abstract – it is using non C strains (and the authors emphasise that there are differences in their in vitro studies), half of the data is previously published. It is useful data, but it needs to be much more clear the limitations. The statistical analysis in 7B is not clear – the red line does not come out clearly is there an R squared value?
5. Where is the TEER data in the first part of figure 1, it would be useful on the side of the fitc assay? Where is the no infection control data for panels 1F, G and H. Why is the bottom right image in 1E at a higher magnification? What are the pink staining in this image – there shouldn't be dsRNA in non infected cells.

Minor

1. Put PGC1a in full in the title, avoids confusion e.g. with Prostaglandin C1.
2. Tightness of language:
 - a. On Page 2 is species the right word for rhinovirus? Serotypes is more accurate
 - b. For the reader's ease, the first time the gene that encodes PGC1 is introduced, explain that it encodes the PGC1 protein.
 - c. Page 7, better to say viral RNA than titres, in the absence of a plaque assay or other infectious assay
 - d. Sentence beginning: If examining HRV on page 8 needs rephrasing as it is unclear.
 - e. Italicise restriction enzymes
 - f. In this line: The AGC and the maximum injection time were set at $1e5$ and 105 ms, respectively. Should the 5 be in superscript? This may not be the case, but it looks like it could be so.
 - g. Page 6, 4th line I think it should say 2B left panel
 - h. Page 4 there is a hanging sentence – starting as neither replication, but doesn't say alternative.
3. In figures – as well as the legends, for ease of the reader would be helpful to say what time points single analyses are performed e.g. 3C
4. SR18292 had a very modest effect in blocking the barrier restoration – statistically significant, but potentially not biologically.
5. The whole paragraph on p14 about gut barrier integrity could be cut, it does not add to the understanding of the study. Replacing with some more information why barrier function in airway infection could be important feels more relevant.
6. Is there any way to measure the virus through an infection assay – counting RNA alone has issues of defective interfering particles – which could be higher in the RV-C prep explaining differences seen.

Response to Reviewers

Please note all changes to the manuscript in response to reviewer suggestions are shown using “Track Changes”

REVIEWER COMMENTS

Reviewer #1 (Remarks to the Author):

I appreciate the opportunity to review this interesting manuscript. As my expertise is metabolomics and proteomics analysis (as well as its integration) in respiratory infection, my review focuses on these areas. As for the validity of experimental models, I humbly defer to the other reviewers. In this study, the authors utilized experimental rhinovirus infection in highly differentiated human bronchial epithelial cells grown at ALI. The authors found that 1) metabolism regulates airway epithelial barrier function, 2) RV-C15 impairs early barrier function prior to viral pattern recognition receptor driven responses through metabolic mechanisms, and 3) PGC-1 α plays a role in regulating barrier function. The authors speculated that PGC-1 α may be a novel therapeutic target for recovering RV-induced barrier loss while decreasing viral titers.

GENERAL COMMENTS

I highly applaud the authors' approaches to use well-thought experimental designs and applications of state of the art metabolomics and proteomics approaches. However, this study has several limitations, such as 1) potential lack of power in these analyses, 2) multiple hypothesis testing, 3) lack of clarity in the statistical analysis, and 4) opportunity to integrate different omics information. Below, I included several comments that might help the authors improve the clarity of the manuscript.

Response

We thank the reviewer's for raising these important points. We fully agree that more information is needed for to provide clarity, and that more integrated analyses add value to the study. While we specifically address comments below in a point-by-point manner, we would like to indicate that we made major changes to the manuscript in regard to new analyses, figures, and methods. We have revised the manuscript with comprehensive detail, and these changed sections are in blue font for the Reviewer to find more easily. As a result of the newly performed and more comprehensive analyses, we draw attention to the addition of a new “Figure 2” for proteomics data and “Figure 3” for metabolomics data. Both Figures 2 and 3 have their respective new Results and Methods in the manuscript. Originally, we did not perform these comprehensive analyses because we are aware of the potential to generate overwhelming amounts of results/data with these methods and we did not want to overwhelm the cell-biology-focused manuscript. However, we believe the new analyses proposed by the Reviewer greatly enhance the depth and insight of our findings and we thank the reviewer for helping us to improve the manuscript.

SPECIFIC COMMENTS

ABSTRACT:

Page 2, para 1: “opportunistic co-infections”. The reviewer is unsure whether RV is a common agent for opportunistic coinfections even in immunocompromised patients. Please clarify or revise this one.

Response

We have changed the wording to read “secondary bacterial infections”. There are numerous references to support that secondary bacterial infections can occur after rhinovirus infections (e.g. Bashir et al, J. Allergy Clin. Immunol. 141:822-824, 2018; Kloepfer et al., J. Allergy Clin. Immunol. 133:1301-1306, 2014; Mallia P et al, Am. J. Respir. Crit. Care Med. 186: 1117-1124, 2012).

INTRODUCTION:

Page 3, para 1: “Although it has been clearly established...”. Please add citation(s) that supports this notion.

Response

Appropriate citations have now been included.

Page 3, para 3: “We employed an unbiased ... proteomics and metabolomics”. The term “unbiased”, while often used, is not appropriate but misleading. “Unbiased” indicates that the point estimate equals to the true value, which is not guaranteed by these testing. Please replace it with a more appropriate word, such as “untargeted”.

Response

We thank the reviewer for pointing this out, and we agree that the word “unbiased” is inappropriate. Our proteomics strategy followed an established bottom-up multiplexed TMT proteomics workflow whereas our metabolomics strategy followed a semi-targeted approach in which 650 compounds were included in our target list, but the underlying methods, high-resolution LC-MS mass spectrometry, are inherently untargeted in nature. Since the word “untargeted” does not adequately capture both workflows, we have removed the term from all portions of the manuscript.

METHODS:

Pages 38-39: The validity of these metabolomics and proteomics measurements is unclear. This is very important information to assess the validity of the study. The authors should report the within-run and between-run CV or the median relative standard deviation for the internal standards.

Response

We understand that providing tables containing the proteomics and metabolomics data is critical to assessing the validity of our findings. We have corrected this initial lack of data availability by performing the following corrections:

Metabolomics: We have now created a Supplementary Document, “Metabolomics Data” containing all of the metabolites that we identified (31 total) for each timepoint and condition for all of the donors. This document contains two tabs:

Tab 1: Donors are organized by condition. The experiment (n=6 donors) was performed simultaneously to minimize experimental variability, meaning we grouped conditions (infected versus non-infected) into their own cell culture plates, further segregated by time point to avoid repeated temperature fluctuations of removing and replacing plates in the incubator which could have affected results. For each plate group (i.e., non-infected controls at 4h) the plate contained one well of PneumaCult Basal Media with no cells to account for natural degradation of media compounds in the incubator. Every plate had a “cell-free media control”, which we extracted with MeOH alongside each donor and analyzed. The cell-free media data can be found alongside the mass spec “blanking” data for the run. We collected each time point, performed the MeOH extractions, and stored the samples at -80C until all of the time points were completed at 24 h. We then ran all of the samples in the same plate in the mass spec in a single run to avoid multi-run plate variability, which can be considerable in LC-MS mass spectrometry.

Tab 2: We present the same metabolite data from Tab 1, now organized by timepoint (non-infected versus infected) and we present the calculated means, medians, standard deviation, and indicate the p value for each two-tailed paired t-test for each metabolite (control versus virus) within each time point. We also include the Benjamini-Hochberg method for FDR, which we calculated using 10% FDR. The adjusted p values are noted along with Benjamini-Hochberg “significance” considering the corrected alpha of 0.039.

Proteomics: We have now deposited our RAW proteomics data into ProteomeXchange via the PRIDE database (accession # PXD024591) and have created a “Data Availability” section of the manuscript containing this information. Additionally, we have provided a Supplementary Proteomics Document “Supplementary Tables 1-7 Proteomics” which includes all RAW data of shotgun proteomics, standard deviation calculations, and each biological replicate TMT 6plex Reporter Intensity. Each subsequent tab is separated by condition and timepoint and lists each protein name and fold change calculation derived from boxplot-and-whiskers analysis, which we described in the improved Methods (Page 47).

Furthermore, the authors should report the number of metabolites and proteins measured / identified. What are the reference libraries to identify / name these metabolites and proteins? Currently, it is difficult to assess the validity of these testing.

Response

We have now added to the manuscript that we identified “2,700 proteins” (Page 6) and “31 known metabolites” (Page 7). In the Methods, we have now added improved descriptions of metabolite identification strategy and libraries (Page 43-44) as well as protein analysis identification and libraries (Page 46-47).

Page 44: The analytical section consists only of four lines, which is highly insufficient. The reviewer has several questions in order to assess the validity of the analysis, such as:

- How was the multiple testing problem addressed, in the setting of sample size of 6?

Response

For metabolomics, we performed 31 paired *t* tests (one for each identified metabolite at each time point). Since we tested numerous hypotheses, we accounted for false discovery rate (FDR) in the chance that we obtained *p* values that were not indeed significant. By applying a Benjamini-Hochberg method, we controlled for false discovery (*i*/*m*)/*Q* where (*p*-value rank/31 tests)*10% FDR. This calculation and corrected alpha values can be found in the Supplementary Table "Metabolomics Data".

- Which software and packages were used (e.g., MSEA with MetaboAnalyst for the pathway analysis)?

Response

For proteomics, we used Metascape and for metabolomics we used MetaboAnalyst. We have updated the manuscript with references to these analysis tools in the Methods (Page 44, 47) and in the appropriate Figure Legends for Figures 2 and 3.

- How were the within-individual/sample correlations addressed?

Response

We used two strategies for addressing potential covariance within samples. The first is that we analyzed metabolites using a univariate strategy and simply ranked our target compounds on the basis of their significance. These data, along with the proteomics data, were also analyzed in MetaboAnalyst, which uses a multivariate strategy for identifying pathways based on co-enrichment of metabolites and proteomics data. These multi-omics analyses were added and presented alongside the univariate analysis strategies. Importantly, the data reported here has been reproduced in multiple independent biological replicates.

- How were the pathway analyses performed? What were the reference libraries?
All of the information is essential to guarantee the reproducibility.

Response

We have amended the manuscript to reflect more detailed descriptions of the analyses and reference libraries used for both proteomics and metabolomics. These changes can be found on Pages 44 (metabolomics) and 46-47 (proteomics), as well as in the Figure Legends for Figures 2 and 3.

RESULTS

Page 5, para 3: This is important. The sample size for the metabolomics and proteomics testing was only 6 while many individual metabolites/proteins as well as pathways were tested (the reviewer speculate that hundreds of hypotheses * multiple time points were tested). This will certainly lead to type 1 errors. How did the author address this critical issue? Please clarify

Response

We agree that Type I Error is a serious problem in the context of multi-omics analyses. We would note, however, that variability of ALI cell culture systems tends to be considerably less than seen with samples collected directly from humans. Thus, while we used a sample size of 30 for our in vivo array studies (Proud et al; Reference 8 of the current paper), we have routinely used lower sample numbers to find significant changes using in vitro culture. Given the significant value we believe these data bring to the field for changing our interpretation of cellular biological processes, we view the multi-omics data as an important resource despite our sample size being only 6. We thus present these data even though they are subject to the same constraints that all multi-omics studies suffer with regard to Type I Error. We note this is consistent with many other published studies using in vitro cultured systems (e.g. Crakes, KR et al, PNAS 116:24819, 2019; Tsai, P-Y et al, PNAS 118:e2003014118, 2021).

Page 6, para 1: Based on the enrichment analysis, the “metabolomic process” is the most significant. However, this is a very generic “process” which consists of numerous “processes”. Can the author comment on this?

Response

We agree with the reviewer and have performed further proteomics analyses to specifically address metabolic pathway associations. Please find these analyses in Figure 2 and Results (Page 5-7). The Methods (Page 47) also reflect these additional analyses.

Additionally, most analyses were performed as a series of cross-sectional analysis. There is an opportunity to examine the longitudinal change in these proteins and metabolites over time after experimental infections.

It is intriguing to see that significant lipid metabolites are substrates of cell membranes (e.g., ethanolamines, L-serine [a substrate of sphingolipids/ceramides]). It may suggest that these metabolites are also results of cell destruction/death instead of increased fatty acid synthesis. The reviewer would like to hear how the authors interpret this important finding.

Response

Although it is always impossible to rule out some small contribution of cell death/turnover, we do not believe this is the main source of metabolites. We base this

on the data in Supplemental Figure 1 where we used two different assays of cell death and showed that, over the first 12 hours, when many of the same metabolites are already being produced (Supplemental Figure 2), cell death was less than 1%. Rather, as mentioned on Page 8 of the revised manuscript, it is well established that picornaviruses, including rhinoviruses (e.g. References 13, 14) remodel and fragment lipid membranes from the endoplasmic reticulum and Golgi to generate replication centers for viral replication to occur. It is well established that alterations in many lipid pathways (including altered cholesterol biosynthesis) are associated with this process. We must also note that membrane changes also likely occur in preparing for extrusion of infected cells from the epithelium.

Pages 6-7: While the sample size may be limited, the study has interesting data – longitudinal measurements of the metabolome, proteome, and six mRNAs. Currently, these data are examined individually and cross-sectionally. There is an interesting opportunity to integrate these data to derive further knowledge. The authors may want to consider additional analyses. There are many techniques/approaches available. Here is a good review article that the might be helpful – Noell et al. Eur Respir Med 2018.

Response

We thank the Reviewer for pointing out the absence of this valuable joint-analysis. We have performed an integrated proteomics and metabolomics analysis (Figure 3e). We also integrated the fold changes from the 6 metabolic-associated genes from Figure 4a; however, adding the gene expression fold changes data had no discernable effect on the joint analysis.

Reviewer #2 (Remarks to the Author):

COMMENTS TO THE AUTHORS:

Common cold infections, caused by human rhinovirus (HRV) are directly associated with exacerbations in those typically with lower airway diseases, including asthma, COPD and CF. Development of vaccines is improbable for these viruses since there are over 150 serotypes known to date and more being identified. As a result, alternative strategies are being sought to combat viral infection. In this study, Michi and colleagues assessed the effects of HRV infection on airway epithelial cell metabolism and associated barrier function. There is very sound premise and execution of this study and the authors should be complemented for such a comprehensive study. Despite this, there are numerous shortfalls that render the manuscript in its current form unsuitable for publication. Firstly, the authors have concentrated on HRV-C which predominantly is associated with exacerbation in young children, however primary cells were derived from adults (only 1- adolescent (13yrs). Secondly, cells were derived from a presumed healthy population whereas the target for the proposed therapies are those with chronic airway diseases whose baselines may be different to those assessed here. Furthermore, the used of uninfected mock controls may not have been the most

appropriate, as the effect viral binding were not thoroughly assessed. Study limitations were not appropriately acknowledged and discussed and there were also several methodological questions that leave the reader unclear about how to reproduce the study.

MAJOR ISSUES:

1. The premise of this paper appears to explore the potential of targeting cellular metabolism and its associated effects on airway epithelial barrier function in response to HRV infection. The rationale behind this is not queried as it has good merit, however, since the target was not age specific there is insufficient clarity as to why the majority of the study solely investigated host effects to HRV-C and not HRV A or B. (as initially perform in the study). This further compounded by the fact that the cohorts used to obtain primary airway cells were all older adults, whereas HRV-C causes pronounced exacerbations in young children. There is evidence that no HRV serotype predominates in adults (McCulloch et al. 2014. Am J Clin Path 142:165) and conflict needs to be addressed.

Response

We do not dispute that there is no HRV species that predominates in adults. Indeed, recent data (Choi et al, Am. J. Respir. Crit. Care Med. – in press) show that neutralizing antibodies to HRV-C species can be found in many older children, at a rate greater than seen for HRV-A species, suggests that pre-existing immunity in older children (and presumably adults) provides protection against severe illness during HRV-C infection with age. However, in *in vitro* culture systems, where neutralizing antibodies (or immune cell-mediated antiviral defenses) are not present, infection with all strains of HRV can still occur regardless of age. We do not have any age or race pre-requisite for lung donors and so receive lungs not used for transplant as available but, in the absence of pre-existing immunity in our model, we believe that they are still appropriate to serve as representative normal human bronchial epithelial cells. Because there is no HRV-species that predominates in adolescents and adults, we initially demonstrated that two strains of HRV-A (each using a different cellular receptor) as well as HRV-C15 **all** caused changes in epithelial permeability, indicating that this is an effect of multiple HRV species/strains. Given that it was impractical to perform all subsequent experiments using multiple strains of HRV, we focused on HRV-C15 for further study because it induced the greatest changes in permeability and so provided the best “signal-to-noise” with which to examine underlying mechanisms. One may reasonably suggest, therefore, that if HRV-C strains also cause the greatest changes in epithelial permeability in young children (with no pre-existing immunity) this could contribute to the more severe disease caused by this species. To further validate that key findings were applicable to multiple strains and species of HRV, however, we also showed (Figure 3C) that the same 3 strains (HRV16, HRV-1A and HRV-C15) **all** caused a reduction in the expression of the gene encoding PGC-1 α . Moreover, their rank order of effect on expression of this gene matched the rank order seen in terms of increasing epithelial permeability. We also showed that, as expected, expression of the gene encoding PGC-1 α was reduced at 48 h post-infection during experimental HRV infections *in vivo* using two different strains of HRV (A16 and A39). As noted in our

manuscript there is no GMP-grade strain of HRV-C approved for clinical use so we could not perform infections with this species. Thus, while we used HRV-C15 for practical reasons in many experiments, we believe that we have established that the key findings are also seen with other strains/species. Since the reduction in effects of HRV-C on symptom severity in older individuals seems to be associated with protection provided by humoral (and perhaps cellular) immunity, the effects of HRV-C in young children likely represents the lack of such systems regulating the actions of HRV-C on epithelial biology.

2. The authors have utilized a specialized medium to grow and induced airway differentiation (PneumCult). However, there are studies suggesting various levels of differentiation obtained with different commercial products (PMID: 32332878, PMID: 32967385) and infer differentiation of primary cells into more the nasal phenotype with this product. The authors will need to justify their differentiation into lower airway cells more strongly in the context of this.

Response

Our lab has been culturing human airway epithelial cells for over 30 years and we are well aware of the complexities of variations in culture medium and culture conditions, as well as of the source of cells used for culture. The airway epithelium of the nasal turbinates and of the trachea/bronchi are both pseudostratified columnar epithelium and are essentially indistinguishable morphologically. We have previously compared several media preparations at ALI and found that culturing primary airway epithelial cells for 5 weeks at ALI using PneumaCult Differentiation Medium gives the best reproduction of *in vivo* airway epithelial morphology. This produces an epithelium that contains basal cells, ciliated and goblet cells, and that recapitulates the structure of *in vivo* epithelium, with dense expression of cilia and consistent mucus production (Warner, et al., *Respir. Res.* 20:150, 2019, and Figure 1A of the current manuscript). It is clear that the morphology of our cultures differs radically from those shown using PneumaCult in the paper by Luengen et al, mentioned by the reviewer (PMID:32332878). This may be because these authors describe using medium with only 10X supplement, which is fine for expansion but is inadequate for differentiation, when basolateral medium should contain 100X supplement, as we use here. In terms of differentiation to a “nasal phenotype” we are unsure what the reviewer is referring to. As noted, nasal and bronchial epithelia are structurally indistinguishable, and a similar frequency of nasal and epithelial cells can be infected with HRV (Mosser AG et al, *J. Infect. Dis.* 185:734-43, 2002). Although there are some papers reporting slightly different levels of cytokine production in response to stimuli between nasal and bronchial cultures, these findings are not seen in other studies and may easily be impacted by varying cell numbers used for seeding, or number of cell divisions between nasal and bronchial cells. By contrast, as we discuss on page 18 of the revised manuscript (with appropriate references) overall gene expression profiles between nasal and bronchial epithelial cells are remarkably consisted. This is also seen upon HRV infection in nasal and bronchial epithelial cells. Thus, the current studies used primary epithelial cells obtained from the 1st to 4th generation bronchi and are

differentiated to a morphology that mimics that seen *in vivo*. Since both nasal and bronchial epithelial cells can be infected with HRV, and show consistent transcriptional profiles, we are confident that our model is the state-of-the-art for studying changes in bronchial epithelial permeability in response to HRV infection.

3. It was difficult to know where appropriate mock controls were used in this study. Since the authors argue, that effects are the result of active infection (of which the experiments were conducted appropriately) the necessary mock infection control to this reviewer would have been UV- inactivated virus. Although some of the generated data does illustrate the inclusion of this type of mock control, the methods suggestion the inclusion of a 'no virus' volume (HEPES based). The reviewer then interpreted the rest of the data to have include this non-viral control. The authors will need to address and interpret their data then in this context, as it's a limitation.

Response

We must respectfully but categorically disagree with the reviewer's assertion that inappropriate controls were used. Our entire study was focused on how HRV infection alters normal epithelial barrier function. As such, it is essential to compare responses in infected cells with those in uninfected cells at the same time points. Since our infection model involved a 2 hour apical exposure of cells to HRV in a small volume of liquid, the best control for this was to expose normal cells to the same volume of the vehicle (F12 with HEPES) that is used as diluent for the virus stock. Both this control and HRV infection samples had this respective liquid removed at the same time point, and then left at ALI with no external liquid above the cells. This system allows us to compare effects of virus infection to non-infected (normal) cells and is standard in the field. This also is the same approach taken for *in vivo* experimental infection studies where a sham (vehicle) exposure of one group of subjects is used to compare to the group receiving HRV infection. It is well established that UV-inactivated virus is still able to bind to its receptor and be internalized but cannot replicate its genome. In the current studies, as in many others we examined the effect of UV-inactivated virus to determine if the alterations in epithelial barrier function upon HRV infection were dependent upon replication (Figure 1C). If UV-inactivated virus reproduced the effects of active HRV, it would imply that we should focus on mechanisms linked to virus-receptor binding and not replication. Since UV-inactivated virus did not cause any alterations in epithelial barrier function relative to control (vehicle exposure as described above) this indicated that replication was required. It is not appropriate, however, to use UV-inactivated virus as a control for normal cells. The airway epithelium *in vivo* would never be exposed to UV-inactivated virus under normal circumstances, and cultured cells exposed to such UV-inactivated virus could not be considered as "normal control cultures".

4. The authors identified transcriptional and translational modifications at the same time point (4 h), which is a very unique finding. Usually post transcriptional modifications occur sometime after transcriptional changes. The authors have not addressed this this observation, just the speed at which observations are made.

Response

Although the reviewer is correct that protein is only produced after some mRNA is transcribed, for numerous genes transcription and translation are often tightly coupled and there are many genes where mRNA and protein induction show similar kinetics. PGC-1 α is a very dynamically controlled molecule at both the transcriptional and translational level (e.g. Adamovic et al. Mol. Cell. Biol.33:2603-2613, 2013). This is perhaps not surprising given the central role of PGC-1 α in regulating numerous metabolic pathways. We now add a comment on page 9 regarding the tight coupling of mRNA and protein.

5. The authors will need to attempt some explanation to the clustering of their cohort (Fig 2B right hand panel). Appears 4 participants clustered and 2 did not. Was this age, gender corrected? Cause of death skewed? This information is needed.

Response

We must confess to being confused by this comment. The original Figure 2B right hand panel showed differences in clustering of ontology clusters of proteomic enriched terms between HRV infected and non-infected donors. There was no individual donor data shown in this panel. We have since removed this panel and replaced it with additional analyses at the request of another reviewer.

6. Fig. 3D should be expanded to include a 24 hr time point so as to be definitive in the return to basal state with Oligomycin A treatment.

Response

The effects of Oligomycin A on barrier function at 24 h is already shown in Figure 6A (right hand panel).

7. The inclusion of experimental infection data into this manuscript is one of the major strengths to this study. Although these prior performed studies were conducted by the same group, the way the manuscript was written infers they were specifically performed as part of this study. The first study was published in 2008 and the second comes from a second as yet unpublished study. The methods section, results, abstract and discussion need to be modified that you fortuitously were able to conduct analyse on these other studies. If the second study is as yet unpublished and the data to be included in this study, the cohort and information needs to become part of this study.

Response

We regret if the reviewer believed we were being misleading in terms of the experimental infection studies. In the Methods section, we explicitly stated that the first study was published in 2008 and that we mined an existing database, even providing the GEO accession number for this database. Similarly, we believed that we clearly indicated in the same Methods section that the second study was ongoing as a comparison of responses to HRV infection between normal subjects and smokers with normal lung function and that we merely mined existing RNAseq data from the normal subjects for expression of PPARGC1a. We have now attempted to clarify this also in the

Results section (page 14). We had already informed the Editorial Office that this ongoing study has been on hold for some time due to the COVID-19 pandemic but that, when complete, it will compare cellular and inflammatory mediator profiles, as well as gene expression profiles and a variety of other parameters between the two populations. Because this study is incomplete, will not overlap with the current manuscript, and because we sampled data only from normal subjects we must respectfully disagree with the reviewer that we should include full cohort datasets.

8. Furthermore, these other corroboration populations samples only nasal epithelium. Although results mirror those found from lower airway epithelial cells which is a strength to the study, there are implications considering the upper and lower airway certainly have very different functions, especially when it comes to pathogen exposure and host responses. The authors will need to address this.

Response

We now clearly state in the Discussion (page 18) that nasal samples were used for *in vivo* studies. We also indicate that studies (reference 46) from others have demonstrated a greater than 90% transcriptional overlap between nasal and bronchial epithelial cells. Moreover, array studies from our lab showed excellent agreement in HRV-induced gene expression profiles between nasal samples and *in vitro* cultured bronchial epithelial cells (references 8 & 47). There is, to date, no clear evidence of any responses to HRV infection that occur in cells from one site and not the other. Thus, while responses to other pathogens could be variable, there is excellent agreement in rhinovirus responses between nasal and bronchial epithelial cells.

9. The discussion lack any comments around those that need the therapeutics the most, ie those with COPD, asthma and CF. There needs to be discussion around the disease cohorts as well as age issues.

Response

We now include a brief discussion of potential cohorts for therapy in the discussion (page 19).

MINOR ISSUES:

1. Introduction page 2-3, second paragraphs, ‘ over 150 serotypes... not species” also ‘three species not genetic groups”

Response

We thank the reviewer for drawing attention to this error. We have changed this in the revised manuscript but have chosen to use the word “strain” rather than serotype, because the latter has a specific meaning. New serotypes of HRV have always been defined based on the fact that neutralizing antibodies to all other known strains failed to block cytotoxic effects of the new virus in appropriate cell lines. There are only about 102 serotypes that have been defined using this method. The additional virus strains (especially the newer HRV-C strains) have been defined based on genetic sequencing.

In addition, there is no cell system to monitor cytotoxic effects of HRV-C strains. Thus, we use strain rather than serotype. We have also corrected groups to species.

2. Page 4 Results “Barrier function disruption was dependent on actively replicating virus, as neither replication deficient UV-treated HRV-C15.” Sentence appears incomplete.

Response

Again, we thank the reviewer for noting this error. The sentence has been changed to “Barrier function disruption was dependent on actively replicating virus, as disruption was not observed using replication deficient UV-treated HRV-C15.”

3. Page 6, Figure 2A left... should be 2B left.

Response

This has been corrected.

4. Page 33: Please state the sources of HRV-1A and 16.

Response

We now indicate in the Methods section (Page 38) that both HRV-1A and HRV-16 were originally obtained from ATCC and provide references for propagation and purification of each strain.

5. Page 33 Please include TEER measurement methodology in appropriate sections. Was this used to corroborate ALI? Add required information in the appropriate sections.

Response

We have now included methodology for measurement of TEER. However, we do not corroborate that ALI cultures are fully differentiated using TEER but using histology.

6. Page 37. Immunoblotting sentence. “Membrane was washed”..... replace with ‘Membranes were washed...’

Response

This has been corrected.

7. Page 37. Probes for SLC2A1 missing (as included in graph 3A) from methods section.

Response

We did state on page 37 (now page 42 of the revised manuscript) that a specific gene expression kit *SLC2A1* (Hs00892681)” was used. This includes primers and fluorescently-labeled probe.

8. Page 38. Include the FITC-Dextran size in methods section (kDa).

Response

The first line of the section headed “FITC-Dextran Permeability” already states that the FITC-Dextran used was 3-5 kDa.

9. Page 41. 3 punch biopsies from the one insert are not considered technical replicates. Inserts will differ in their differentiation and their infectivity and repeated inserts are required for these measurements (oxygen consumption quantification).

Response

We have removed the wording of technical replicates and replaced with “three measurements per insert”. We note that the inserts used to grow our ALI cultures are 1.12 cm² in area. A characteristic of HRV infections *in vivo* is that they do not evenly infect all epithelial cells but rather cause a “patchy” infection that has been described in multiple papers. Consistent with this, we have observed, using dsRNA staining, that infection of epithelial cells in our ALI cultures is also not uniform across the insert. To get the best sampling for oxygen consumption measurements, therefore, we perform punch biopsies in three to four regions of each insert. Thus, each data point in Figures 4 B&C is the average of the 3-4 measurements from that insert and a total of 3 to 4 different lung donors were used for each condition. Thus 9-12 total punches were used for each experimental condition, depending on the panel, as explained in the Methods and Figure legends. We would note that we do not see variations in differentiation status or infectivity between inserts from within each donor. The major variation occurs between donors, which is reflected in the distribution of data points within Figures 4 B&C.

10. Call confocal images will need to be addressed. It appears that they were counterstained with DAPI but it is extremely difficult to visualize this in many panels. All panels need to be normalized for both TJ and DAPI.

Response

ALI cultures recapitulate the morphology of *in vivo* pseudostratified columnar epithelium extremely well (Figure 1a: histological sections). Thus, the ALI cultures are around 50-70 μ M in height. We understand that typically immunofluorescence/confocal microscopy is performed on monolayers of cells in which the cells and their respective nuclei are in the same focal plane (due to uniform height). However, columnar epithelial cells have nuclei that are situated closer to the basal cells, which means that when stained for DAPI (blue), they are not in the same focal plane as the apically expressed tight junction proteins (i.e., ZO-1/occludin), which are situated directly beneath cilia. In non-infected cells, the nuclei cannot be seen (as seen in Figure 1a: non-infected or Figure 7d: non-infected). However, as HRV infected cells begin to display the cytoskeletal rearrangement-driven effects of HRV infection, the columnar ciliated cells are extruded from the epithelium and extruded cell nuclei co-localize with apically expressed tight

junction proteins. This is why there is no nuclei (DAPI) staining visible in non-infected panels. To confirm these morphological alterations seen in immunofluorescence/confocal microscopy, we performed immunohistochemistry (the gold standard for morphological assessment) (Figure 1a bottom row) to confirm that infected cells are indeed being extruded (Figure 1a HRV-1A and HRV-C15 histological sections).

11. Figure 1E . Non-infected dsRNA panel scale (25um) not equivalent to rest of panel (50um). Please replace with appropriate panel.

Response

This has been corrected.

12. Supplement Figure 1D UVHRVC15 panel again different scale. Replace.

Response

This has been corrected.

Reviewer #3 (Remarks to the Author):

In the current study, the authors look at the impact of RhinovirusC15 on barrier function in ALI models. They demonstrate that infection induces a metabolic remodelling that contributes to the disintegration of barrier function. Pharmacological intervention in this process can reduce viral replication.

Major

1. The generalisation of this finding – is it restricted to just one Rhinovirus isolate? Could you confirm in another C virus? But is it more broad, many viruses disrupt barrier function, is it unique to rhinovirus?

Response

In Figure 1A and 1B we demonstrated that three different strains of HRV (HRV-16, HRV-1A and HRV-C15), each of which use a different receptor to gain cell entry, all had effects on epithelial barrier function, as assessed both by FITC-dextran movement and by staining of tight junction proteins. Although all 3 strains caused some barrier disruption, there was a rank order effect with HRV-16 being least effective, HRV-1A being intermediate and HRV-C15 exerting the greatest effect. In Figure 3C we also showed that the same 3 virus strains also all modulated expression of the gene encoding PGC-1 α . The extent of gene regulation also varied with each of the strains but, importantly, the rank order of effect mirrored the effects of the 3 strains on barrier function. Our studies of gene expression from *in vivo* experimental infection studies verified downregulation by HRV-16 and showed that HRV-39 also downregulated the gene encoding PGC-1 α . Thus, we have clearly shown that our main findings are not limited to a single rhinovirus isolate. We did not extend to other respiratory virus types

in the current study in order to maintain our focus on further mechanistic studies in the paper.

2. Many of the introductory statements leading into a block of work could be cut as they feel either repetitive (referencing early parts of the manuscript) or over-interpretive e.g. 'Given that changes in barrier function were not dependent upon conventional host innate antiviral signaling' on p5. This statement is not supported, it is inference from the timing of response.

Response

Although we believe that some statements are needed to establish the rationale for the studies that then follow. We have gone through the manuscript to try to reduce the number of statements used in this manner.

3. Much of the text/ discussion emphasises the novelty/ first that this work has been done – this is unnecessary and potentially inaccurate. For example Unger et al examined ROS on barrier function in RV infection (with a different strain of RV: <https://pubmed.ncbi.nlm.nih.gov/24429360/>). Would be clearer to state the results and let the reader draw their own interpretation of the impact/ importance.

Response

We have read through the manuscript carefully and find few if any statements reporting to be “the first” to conduct things. If the reviewer has specific examples, we will be pleased to review them. We did state that the relationship between mitochondrial metabolism and epithelial barrier function has not previously been studied “*in highly differentiated airway epithelial cultures*”. This was a simply a statement of fact. We are well aware of the paper by Unger and colleagues. Their work was performed using the 16HBE cell line, which, unlike primary epithelial cells, does not lead to a differentiated epithelium at ALI. Moreover, their examination of mitochondrial ROS was conducted at 16 h after infection, which is far later than when we see epithelial barrier alterations.

4. More clarity on the use of human challenge needs to be made in the abstract – it is using non C strains (and the authors emphasise that there are differences in their *in vitro* studies), half of the data is previously published. It is useful data, but it needs to be much more clear the limitations. The statistical analysis in 7B is not clear – the red line does not come out clearly is there an R squared value?

Response

Unfortunately, we are already at the word limit for abstracts for articles in *Nature Communications*. As noted in our response to point 1, we have shown that three different strains of HRV, all using different receptors alter epithelial barrier function and modulate expression of PGC-1 α . As we note on page 14 of the manuscript, there is no preparation of HRV-C that is approved for human use, therefore we were limited to the GMP-grade strains approved by the FDA and by Health Canada. Nonetheless, the data obtained are consistent with our *in vitro* data that multiple strains of HRV can regulate expression of the gene encoding PGC-1 α . It is true that we mined an existing dataset

looking specifically for this gene but we explicitly stated that the first study was published in 2008 and that we mined an existing database, even providing the GEO accession number for this database. We have now attempted to clarify this also in the Results section (page 14). We now also provide more detail on the analysis method used, which is common for datasets with multiple time points. We fit a linear model using the formula: $\log_2(\text{PPARGC1A counts}) \sim \text{subject} + \text{study day}$. Study day was encoded as an ordered factor, and we tested the coefficient for the linear trend. The formula used is analogous to carrying out a paired t-test if we were simply comparing e.g. day 0 to day 3 (i.e. we're asking whether the slope of the change over time is non-zero). We have now included this in the Figure Legend and statistical methods section.

5. Where is the TEER data in the first part of figure 1, it would be useful on the side of the fitc assay? Where is the no infection control data for panels 1F, G and H. Why is the bottom right image in 1E at a higher magnification? What are the pink staining in this image – there shouldn't be dsRNA in non infected cells.

Response

We are unclear as to which graph the reviewer is specifically referring to. There are three functional FITC-dextran assays shown in Figure 1. Figure 1B shows the HRV strain comparison at 24h. Figure 1C shows Poly IC, UV, and live HRV at 12 and 24h. Figure 1F shows a comprehensive time course of HRV-C15 specifically. The only experiment where we measured TEER is Figure 1F, in which the TEER data is plotted directly underneath the FITC-dextran time course graph (now Figure 1g) for optimal visual comparison.

In the original Figure 1G (now Figure 1e), we did not plot the non-infected control intracellular HRV-C15 because there is no detectable viral RNA in those cells, as they received a mock solution. In Figure 1H, we also did not plot levels of IFN mRNA in non-infected cells because the levels are below the detection limit of the RT-PCR as non-infected cells do not show detectable changes in IFN-lambda or IFN- beta mRNA.

In Figure 1E (now Figure 1i), we performed a higher magnification of the dsRNA panel from infected cells to show the perinuclear localization of dsRNA. The figure legend states that dsRNA is "pink" yet we understand that the way the confocal panel is arranged that this may be confusing. We have corrected the panel layout to make this more clear.

Minor

1. Put PGC1a in full in the title, avoids confusion e.g. with Prostaglandin C1.

Response

Nature Communications limits the length of Titles to 15 words. Thus, we could not use the full name for PGC-1 α .

2. Tightness of language:

a. On Page 2 is species the right word for rhinovirus? Serotypes is more accurate

Response

We thank the reviewer for noting this error. For reasons described under responses to Reviewer 2 we have replaced “species” with “strains” rather than serotypes.

b. For the reader’s ease, the first time the gene that encodes PGC1 is introduced, explain that it encodes the PGC1 protein.

Response

We have now done this on page 9, first line.

c. Page 7, better to say viral RNA than titres, in the absence of a plaque assay or other infectious assay

Response

“Titers” has now been corrected to “intracellular RNA copy number”.

d. Sentence beginning: If examining HRV on page 8 needs rephrasing as it is unclear.

Response

This has been done as requested.

e. Italicise restriction enzymes

Response

This has been corrected.

f. In this line: The AGC and the maximum injection time were set at 1e5 and 105 ms, respectively. Should the 5 be in superscript? This may not be the case, but it looks like it could be so.

Response

This has been corrected.

g. Page 6, 4th line I think it should say 2B left panel

Response

This has been corrected.

h. Page 4 there is a hanging sentence – starting as neither replication, but doesn’t say alternative.

Response

Again, we thank the reviewer for noting this error. The sentence has been changed to “Barrier function disruption was dependent on actively replicating virus, as disruption was not observed using replication deficient UV-treated HRV-C15.”

3. In figures – as well as the legends, for ease of the reader would be helpful to say what time points single analyses are performed e.g. 3C

Response

In each figure legend we have indicated the timepoints which reflect each data set. Figures representing time courses are labeled such that the X axis is time. Figures performed at a single timepoint are all indicated in the figure legends. We respectfully disagree that adding these to panels as well would be beneficial.

4. SR18292 had a very modest effect in blocking the barrier restoration – statistically significant, but potentially not biologically.

Response

Although the magnitude of the effect of SR18292 on phenotype was more modest than that seen with the PGC-1 α activator ZLN005, the effect seen with SR18292 provides one more piece of evidence that is consistent with the overall finding that PGC-1 α regulates epithelial barrier function. A likely explanation for the lower magnitude of effect is that oligomycin A, even at low concentrations, was too potent and effective for SR18292 to more strikingly overcome oligomycin A's effects on PGC-1 α induction or barrier recovery reversal. Nonetheless the data add one more piece of evidence to support the role of PGC-1 α in regulating barrier function.

5. The whole paragraph on p14 about gut barrier integrity could be cut, it does not add to the understanding of the study. Replacing with some more information why barrier function in airway infection could be important feels more relevant.

Response

We must respectfully disagree with the reviewer. Because there is essentially minimal data from airways linking cellular metabolism and barrier function, the gut epithelium provides the only precedent for the regulation of barrier function and metabolism at a mucosal surface by viruses and, as such, we strongly believe it is important to discuss this for important context.

6. Is there any way to measure the virus through an infection assay – counting RNA alone has issues of defective interfering particles – which could be higher in the RV-C prep explaining differences seen.

Response

Unfortunately, this is a limitation in the current state-of-the art for HRV-C species. There is no current cell line system in which HRV-C species cause any form of cytopathic effect to permit a plaque assay or a measurement of TCID₅₀. Indeed, Dr. Yury Bochkov who identified CDHR3 as the receptor for HRV-C strains, created a HeLa cell line that expresses CDHR3 hoping that this HeLa-E8 cell line could be used for

plaque assays. However, while expression of CDHR3 rendered HeLa-E8 susceptible to infection by HRV-C strains it was not susceptible to cell lysis, which is a critical requirement for plaque assays.

REVIEWER COMMENTS

Reviewer #1 (Remarks to the Author):

I appreciate the authors' efforts to address many comments. My comments have been adequately addressed and the quality/clarity of manuscript has improved. I have one minor suggestion -- in somewhere in the Discussion section, the authors should acknowledge that a detailed longitudinal and integrated omics analysis was not performed on the cell-biology focused manuscript and that this needs to be done in another form of investigation.

Reviewer #2 (Remarks to the Author):

The revised manuscript, edited data and clarified text now have addressed all initial critiques raised by this reviewer. The resulting manuscript is now acceptable for publication in my opinion making significant contribution to the field.

Reviewer #3 (Remarks to the Author):

The new data looks good thanks for including, there are still some presentation issues from the initial review that I would like to see incorporated.

1. Some discussion on limitations. These findings apply to a single strain of virus. Further work will be required to confirm in other HRV viruses.
2. In figure 1, sorry if the comment was unclear. Figure 1e-h need to have the negative control data INCLUDED in the figure. Even if it shows negative data. People reading the paper will not know that these controls have been performed unless the data is included. Can be put at the limit of detection for each timepoint.
3. In figure 1. For the TEER, I am suggesting to include additional data to panel 1b to show the impact of infection on this, not asking where missing data was.
4. The answer in the rebuttal about the Unger paper was very clear, please include in the actual manuscript to clarify the differences with what has gone before.
5. The authors asked for incidences of 'the first time', Incidences of First: Line 363, 392, 426. Have never been studied: Line 239
6. The abstract and word count: The point about the human challenge data is that it is quite skinny. It is re-derived RNA transcriptomics of a single gene (panel A) and PCR of a single gene (Panel B). Both data sets were derived using a different virus to the majority of the study. The abstract suggests that it was used to explore barrier function. If the authors are concerned about word count, the simplest answer is to remove the words 'human challenge' from the abstract.

Minor

Line 169 – suggest were not statistically significant rather than did not reach. And then remove the speculation as to why they were not significant as this doesn't add anything.

Line 361: double negative. Least effective in loss. Better say caused the least

Response to Reviews

REVIEWER COMMENTS

Reviewer #1 (Remarks to the Author):

I appreciate the authors' efforts to address many comments. My comments have been adequately addressed and the quality/clarity of manuscript has improved. I have one minor suggestion -- in somewhere in the Discussion section, the authors should acknowledge that a detailed longitudinal and integrated omics analysis was not performed on the cell-biology focused manuscript and that this needs to be done in another form of investigation.

RESPONSE

We have now included an acknowledgement that a detailed longitudinal and integrated omics analysis was not performed on page 17 of the revised manuscript.

Reviewer #2 (Remarks to the Author):

The revised manuscript, edited data and clarified text now have addressed all initial critiques raised by this reviewer. The resulting manuscript is now acceptable for publication in my opinion making significant contribution to the field.

NO RESPONSE REQUIRED

Reviewer #3 (Remarks to the Author):

The new data looks good thanks for including, there are still some presentation issues from the initial review that I would like to see incorporated.

1. Some discussion on limitations. These findings apply to a single strain of virus. Further work will be required to confirm in other HRV viruses.

RESPONSE

We now state on page 19 of the Discussion that demonstrating that modulation of PGC-1 α activity regulated barrier function has only been done using infection with HRV-C15 and that further studies are needed to extend this to additional strains.

2. In figure 1, sorry if the comment was unclear. Figure 1e-h need to have the negative control data INCLUDED in the figure. Even if it shows negative data. People reading the paper will not know that these controls have been performed unless the data is included. Can be put at the limit of detection for each timepoint.

RESPONSE

We now include a revised Figure 1 that includes negative control data as requested. For these figures, we also provided statements in each Figure Legend that HRV (or antivirals) was not detected in non-infected ALI cultures. We are also providing all of the raw data for each figure, as required for *Nature Communications* publications, which will show all "undetected" data values.

3. In figure 1. For the TEER, I am suggesting to include additional data to panel 1b to show the impact of infection on this, not asking where missing data was.

RESPONSE

We are unable to show TEER as we did not perform measurements of TEER in the studies shown in Figure 1B. There were several reasons for this. As shown in Figure 1g, TEER is a less sensitive index of barrier disruption than FITC-dextran passage, as can be seen at early time points when there are no significant changes in TEER despite significant passage of FITC-dextran. Moreover, having shown disruption of barrier using functional FITC-dextran movement, disruption of two different tight junction proteins, and immunohistochemistry (Figure 1a) an additional parameter would be redundant. Finally, by 24 h, much of the loss of barrier is due to extrusion of infected cells (Figure 1i) and TEER becomes less reliable. It would take several months to grow new ALI cultures to repeat these experiments with, in our opinion, no change in the meaning of experiments.

4. The answer in the rebuttal about the Unger paper was very clear, please include in the actual manuscript to clarify the differences with what has gone before.

RESPONSE

We now include this on page 10 of the paper.

5. The authors asked for incidences of 'the first time', Incidences of First: Line 363, 392, 426. Have never been studied: Line 239

RESPONSE

Each of these instances has now been changed to remove the use of "first"

6. The abstract and word count: The point about the human challenge data is that it is quite skinny. It is re-derived RNA transcriptomics of a single gene (panel A) and PCR of a single gene (Panel B). Both data sets were derived using a different virus to the majority of the study. The abstract suggests that it was used to explore barrier function. If the authors are concerned about word count, the simplest answer is to remove the words 'human challenge' from the abstract.

RESPONSE

As suggested, we have removed the words "in humans" from the abstract.

Minor

Line 169 – suggest were not statistically significant rather than did not reach. And then remove the speculation as to why they were not significant as this doesn't add anything.

RESPONSE

Corrected as requested.

Line 361: double negative. Least effective in loss. Better say caused the least

RESPONSE

Corrected as requested.